# Transient cell-in-cell formation underlies tumor relapse and resistance to immunotherapy

Amit Gutwillig[1], Nadine Santana-Magal[1], Leen Farhat-Younis[1], Diana Rasoulouniriana[1], Asaf Madi[1], Chen Luxenburg[2], Jonathan Cohen[2], Krishnanand Padmanabhan[2], Noam Shomron[2], Guy Shapira[2], Annette Gleiberman[1], Roma Parikh[3], Carmit Levy[3], Meora Feinmesser[1,4], Dov Hershkovitz[1,5], Valentina Zemser-Werner[5], Oran Zlotnik[6], Sanne Kroon[7], Wolf-Dietrich Hardt[7], Reno Debets[8], Nathan Edward Reticker-Flynn[9], Peleg Rider[1], Yaron Carmi[1]*

[1]Department of Pathology, Sackler School of Medicine, Tel Aviv University, Tel Aviv, Israel; [2]Cell and Developmental Biology, Sackler School of Medicine, Tel Aviv University, Tel Aviv, Israel; [3]Human Molecular Genetics and Biochemistry, Sackler School of Medicine, Tel Aviv University, Tel Aviv, Israel; [4]Institute of Pathology, Rabin Medical Center- Beilinson Hospital, Petach Tikva, Israel; [5]Institute of Pathology, Tel Aviv Sourasky Medical Center, Tel Aviv, Israel; [6]Department of General Surgery, Rabin Medical Center- Beilinson Campus, Petach Tikva, Israel; [7]Department of Biology, Institute of Microbiology, Zurich, Switzerland; [8]Department of Medical Oncology, Erasmus MC Cancer Institute, Rotterdam, Netherlands; [9]School of Medicine, Department of Pathology, Stanford University, Stanford, United States

*For correspondence:
yaron.carmi@gmail.com

**Competing interest:** The authors declare that no competing interests exist.

**Abstract** Despite the remarkable successes of cancer immunotherapies, the majority of patients will experience only partial response followed by relapse of resistant tumors. While treatment resistance has frequently been attributed to clonal selection and immunoediting, comparisons of paired primary and relapsed tumors in melanoma and breast cancers indicate that they share the majority of clones. Here, we demonstrate in both mouse models and clinical human samples that tumor cells evade immunotherapy by generating unique transient cell-in-cell structures, which are resistant to killing by T cells and chemotherapies. While the outer cells in this cell-in-cell formation are often killed by reactive T cells, the inner cells remain intact and disseminate into single tumor cells once T cells are no longer present. This formation is mediated predominantly by IFNγ-activated T cells, which subsequently induce phosphorylation of the transcription factors signal transducer and activator of transcription 3 (STAT3) and early growth response-1 (EGR-1) in tumor cells. Indeed, inhibiting these factors prior to immunotherapy significantly improves its therapeutic efficacy. Overall, this work highlights a currently insurmountable limitation of immunotherapy and reveals a previously unknown resistance mechanism which enables tumor cells to survive immune-mediated killing without altering their immunogenicity.

## Editor's evaluation

This is a timely and important study that describes a new potential mechanism of resistance to immune checkpoint blockade. Not only does this have significant implications for cancer immunotherapy, but could extend to other immunological malignancies as well.

**eLife digest** Cancer immunotherapies use the body's own immune system to fight off cancer. But, despite some remarkable success stories, many patients only see a temporary improvement before the immunotherapy stops being effective and the tumours regrow.

It is unclear why this occurs, but it may have to do with how the immune system attacks cancer cells. Immunotherapies aim to activate a special group of cells known as killer T-cells, which are responsible for the immune response to tumours. These cells can identify cancer cells and inject toxic granules through their membranes, killing them. However, killer T-cells are not always effective. This is because cancer cells are naturally good at avoiding detection, and during treatment, their genes can mutate, giving them new ways to evade the immune system. Interestingly, when scientists analysed the genes of tumour cells before and after immunotherapy, they found that many of the genes that code for proteins recognized by T-cells do not change significantly. This suggests that tumours' resistance to immune attack may be physical, rather than genetic.

To investigate this hypothesis, Gutwillig et al. developed several mouse tumour models that stop responding to immunotherapy after initial treatment. Examining cells from these tumours revealed that when the immune system attacks, they reorganise by getting inside one another. This allows some cancer cells to hide under many layers of cell membrane. At this point killer T-cells can identify and inject the outer cell with toxic granules, but it cannot reach the cells inside.

This ability of cancer cells to hide within one another relies on them recognising when the immune system is attacking. This happens because the cancer cells can detect certain signals released by the killer T-cells, allowing them to hide. Gutwillig et al. identified some of these signals, and showed that blocking them stopped cancer cells from hiding inside each other, making immunotherapy more effective.

This new explanation for how cancer cells escape the immune system could guide future research and lead to new cancer treatments, or approaches to boost existing treatments. Understanding the process in more detail could allow scientists to prevent it from happening, by revealing which signals to block, and when, for best results.

## Introduction

Given the direct correlation between the prevalence of cytotoxic T-cell immunity and improved clinical outcomes (*Pagès et al., 2005*; *Galon et al., 2006*; *Fridman et al., 2012*), most therapeutic strategies are aimed at harnessing T-cell immunity to fight cancer. These attempts include de novo expansion of tumor-infiltrating cytotoxic T cells (*Restifo et al., 2012*; *Rosenberg, 2014*), engineered T cells (*Gross et al., 1989*; *Kalos et al., 2011*), or blocking antibodies directed against suppressive receptors (*Pardoll, 2012*; *Hodi et al., 2010*). However, long-term follow-up indicates that while patients will experience tumor regression, it is frequently followed by recurrence of tumors that are largely resistant to subsequent treatments (*Salati et al., 2018*; *Gide et al., 2018*; *Gauci et al., 2019*). As tumor cells often express mutated proteins or neo-antigens (*Vogelstein et al., 2013*; *Schumacher and Schreiber, 2015*), it remains unclear why eliciting the full spectrum of T-cell immunity to these antigens is not sufficient to eradicate tumors.

Several theoretical frameworks have been widely employed to understand how tumors escape T-cell-based therapies. The main conceptual paradigm suggests that in order to avoid killing by T cells, tumor cells edit or lose their targeted antigens, or downregulate HLA molecules (*Khong and Restifo, 2002*; *Dunn et al., 2002*; *Gajewski et al., 2013*). However, the presence of immunogenic clones expressing neoantigens and HLA molecules, despite having infiltrated reactive T cells, has been well-documented (*Straetemans et al., 2015*). Accordingly, tumor resistance to immunotherapy was also shown to be acquired through tumor cell-intrinsic mechanisms, including loss-of-function of JNK and PTEN, MYC overexpression, and constitutive WNT signaling (*Zaretsky et al., 2016*; *Sade-Feldman et al., 2017*; *Peng et al., 2016*). Alternative explanation regarding how immunogenic clones can survive T-cell pressure is attributed to the capacity of relapsed tumors to promote a suppressive microenvironment (*Sharma et al., 2017*; *Topalian et al., 2015*). Along these lines, the discovery that tumor cells, as well as tumor-infiltrating cells, actively express receptors that negatively regulate T cells (i.e. PD-L1, PD-L2, and etc) activity highlighted novel mechanisms by which tumors

escape immunosurveillance (*Dong et al., 2002*; *Iwai et al., 2002*). Nonetheless, only about 30% of the patients achieve durable objective response following treatment with checkpoint blockade (*Topalian et al., 2015*; *Egen et al., 2020*; *Haslam and Prasad, 2019*). Moreover, combination therapies of tumor-reactive T cells with immunomodulators such as regulatory receptors blockades are still not sufficient to profoundly improve response rates (*Rohaan et al., 2018*; *Swart et al., 2016*), which strongly suggests that additional escape mechanisms are involved.

While these frameworks provide important insights into the complex interactions between reactive T cells and tumor cells, why tumors recur in patients undergoing complete response following immunotherapy remains largely unknown. Most of our knowledge on this process comes from experiments comparing individuals who respond or do not respond to immunotherapy, and not from experimental settings that measure the cell-intrinsic changes, neoantigen landscape, and immune parameters within the same individual. Surprisingly, whole exome and genome sequencing comparisons of paired primary and relapsed tumors indicate that most clones in relapsed tumors are shared with the primary tumors, suggesting relatively low rates of clonal deletion (*Yates et al., 2017*; *Hao et al., 2016*; *Gundem et al., 2015*).

Therefore, to study this process in melanoma and breast cancer, we developed several mouse models in which resistant tumors relapse following immunotherapies. In both mice and humans, we found that the tumor cells remaining after immunotherapy form unique cell-in-cell structures and generate membrane architecture that is impenetrable by immune-derived lytic granules, cytotoxic compounds, and chemotherapies. While the outer cells in this formation are often killed by reactive T cells, the inner cells maintain their integrity and viability and can survive for weeks in culture containing reactive T cells. Once T cells are removed, the inner tumor cells disseminate back into their parental single cells. Overall, this work suggests that this biological process is a central mechanism through which tumor cells escape T-cell immunity and give rise to relapsed tumors.

## Results

### Relapsed tumor cells share neoantigens with primary tumors

To study tumor relapse, we initially looked for a mouse model in which tumors recur following their complete regression. Since the effect of classic checkpoint blockade therapies are limited in most mouse models, we used a combination of dendritic cell adjuvant with tumor-binding antibodies to elicit highly potent T-cell immunity (*Spitzer et al., 2017*; *Carmi et al., 2015*). A version of this therapy is now tested in a multi-center phase-I trial (*Ackerman et al., 2021*). While this treatment does not recapitulate the exact effect of checkpoint blockade therapy, it induces T cell-dependant immunity that completely eradicates established tumors across multiple models. Therefore, when palpable melanoma tumors were injected with a combination of TNFα, CD40L, and an antibody against the melanoma antigen TRP1, tumors were almost completely eradicated in all mice. Nonetheless, after about 10 days, half the mice developed recurrent tumors that were resistant to subsequent treatments (*Figure 1A*, *Figure 2—video 1*; *Figure 1—figure supplement 1*). Similar patterns of tumor relapse and resistance were also observed upon treating BALB/c mice bearing breast tumors with allogeneic antibodies and DC adjuvant (*Figure 1A*). To test if this occurrence represents a broader phenomenon, we also treated melanoma-bearing mice with splenic CD8[+] T cells expressing TCR against gp100 or TRP2 melanoma antigens. Both treatments induced significant tumor regression, followed by tumor recurrence in all treated mice (*Figure 1B*, *Figure 1—figure supplement 1B*). As observed with the previous immunotherapy, recurrent tumors were resistant to subsequent treatment with gp100- or TRP2-reactive T cells (*Figure 1B*). In all cases, immune composition analysis of tumors following the second immunotherapy indicated treated mice across all immunotherapies show massive immune-cell and T-cell infiltration (*Figure 1C*) with a dominant representation of T-cell clones against the melanoma antigens TRP2 and gp100 (*Figure 1D*). To test if these tumor-infiltrating T cells (TIL) are active, we adoptively transferred them to naive mice and challenged them with tumor inoculation. In two different tumor models, TIL transfer prevented tumor growth, further indicating that their overall activity favours immunity (*Figure 1E-F*). Next, we assessed changes in immunogenicity of tumors following immunotherapy. Analysis of melanoma cells showed they maintain expression of MHC-I (*Figure 1—source data 1*), as well as TRP2 and gp100 antigens (*Figure 1G*, *Figure 1—figure supplement 1*). To assess changes in immunogenicity of resistant cell lines, beyond TRP2 and gp100

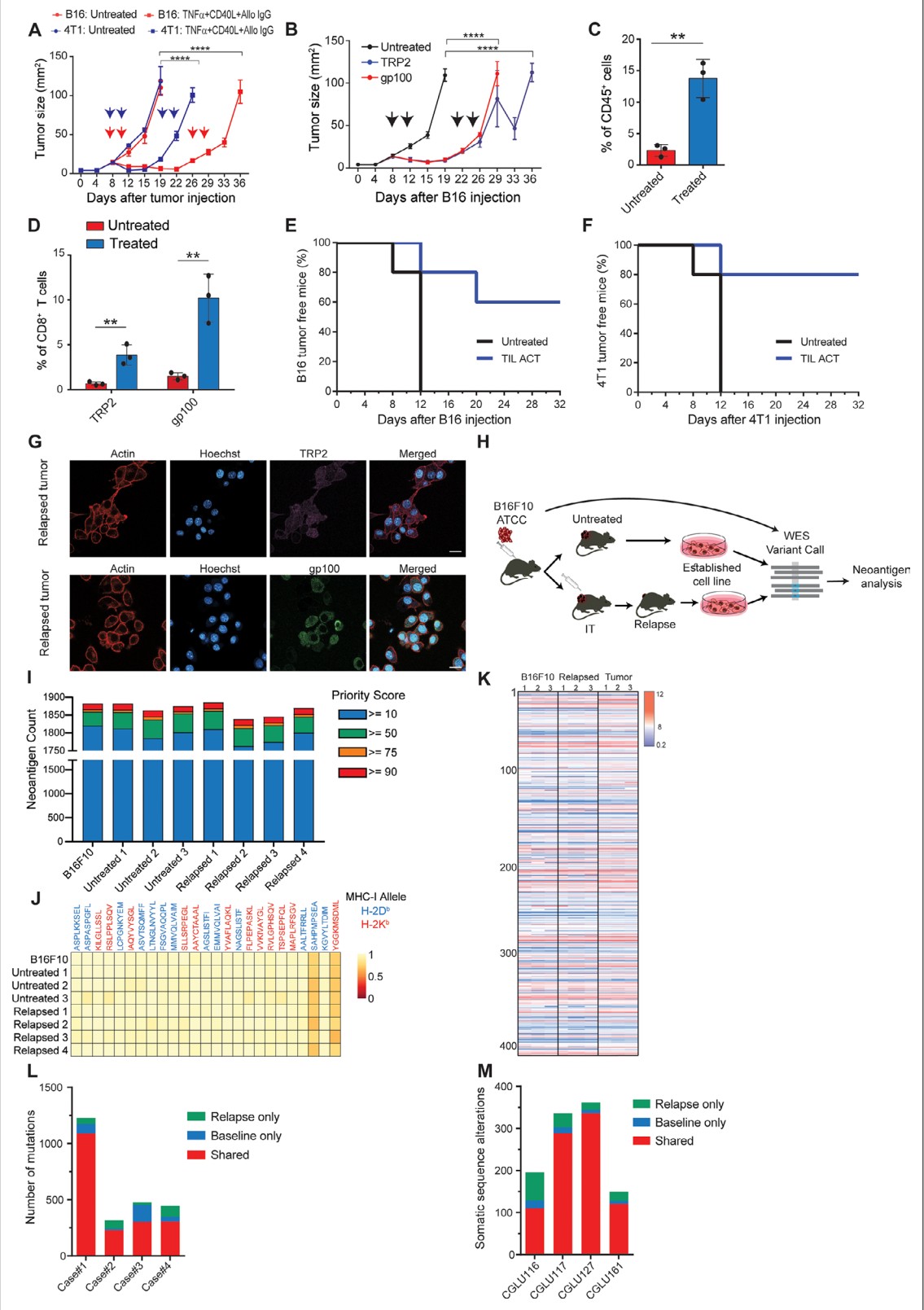

**Figure 1.** Immunotherapy-relapsed tumor cells share neoantigens with primary tumors. (**A**) Tumor size (in mm²) following treatments with anti-CD40, TNFα(i.e. (DC) adjuvant) and tumor-binding antibodies (n=5). Arrowheads indicate treatment. (**B**) Tumor size (in mm²) following adoptive transfer of gp100-reactive or TRP2-reactive CD8⁺ T cells (n=5). Arrowheads indicate treatment (**C**) Mean percentages of T cells out of CD45⁺ cells in B16F10 tumors treated with DC adjuvant and tumor-binding antibodies (n=3). (**D**) Mean percentage of TRP2- and gp100-reactive T-cell clones in tumors treated with

*Figure 1 continued on next page*

*Figure 1 continued*

DC adjuvant and tumor-binding antibodies (n=3). (**E**) Percentage of tumor-free mice injected with CD8$^+$ T cells from immunotherapy-treated B16 tumors (ACT), and subsequently challenged with B16F10 cells (n=5). (**F**) Percentage of tumor-free mice injected with CD8$^+$ T cells from immunotherapy-treated 4T1 tumors (ACT), and subsequently challenged with 4T1 cells (n=5). (**G**) Representative staining of TRP2 and gp100 in immunotherapy- relapsed B16F10 cells. (**H**) Illustration of neoantigen discovery pipeline. (**I**) Neoantigen burden in B16F10 cells isolated from untreated tumors, or following immunotherapy-treated B16F10 tumors (IT). (**J**) Allele frequency comparison of 25 neoantigens with the highest MHC-I affinity. (**K**) RNAseq expression level of B16F10-known neoantigens. (**L**) Primary and relapsed tumors non-synonymous mutation load in melanoma patients (n=4). (**M**) Somatic sequence alterations in NSCLC Patients (n=4). Experiments were repeated independently at least three times. Statistical significance was calculated using ANOVA with Tukey's correction for multiple comparisons (** denotes p<0.01, **** denotes p<0.0001). Error bars represent standard error. Scale bars = 20 µm.

The online version of this article includes the following source data and figure supplement(s) for figure 1:

**Source data 1.** Log2 expression of B16F10 neoantigens, related to *Figure 1*.

**Figure supplement 1.** Comparing the immunogenicity of primary and relapsed tumors, Related to *Figure 1*.

expression, we analysed their neoantigen landscape in comparison to that of B16F10 from untreated mice. To this end, we established four lines of B16F10 that relapsed following immunotherapy and were resistant to subsequent treatments (*Figure 1H*). Whole exome analysis (WES) indicated comparable neoantigen burden across all samples (*Figure 1I*) with the same 25 germline mutations with the highest MHCI affinity (*Figure 1J*). RNAseq analyses further demonstrated similar patterns of gene expression (*Figure 1—figure supplement 1H*) as well as known B16F10 neoantigens in cell lines established from relapsed tumors sorted following effective immunotherapy (*Figure 1K*, *Figure 1—source data 1*). To assess the human relevance of these findings, we next plotted the neoantigen burden identified in relapsed melanoma (*Zaretsky et al., 2016*) and in non-small cell carcinoma patients *Anagnostou et al., 2017* following treatment with checkpoint blockade. Consistent with our finding in mice, the majority of neoantigens were shared between relapsed tumors and their corresponding primary tumor (*Figure 1L–M*).

## Tumor cells surviving immunotherapy organize in a transient cell-in-cell formation

To assess whether tumors acquire resistance through inherent mechanisms, we next tested the susceptibility of cell lines established from treated mice to subsequent immunotherapy. Initially, cell lines established from tumors relapsed following TNFα, CD40L and anti-TRP1 were injected to naïve mice and allowed to reach a palpable size. Consistent with their patterns of gene and neoantigens expression, these tumors were equally susceptible to the same immunotherapy as the parental B16F10 cells, albeit relapsed more rapidly (*Figure 2A*, *Figure 2—figure supplement 1A*). Interestingly, these cell lines maintained their baseline levels of MHC-I, TRP2 and gp100 proteins albeit encountered with gp100 and TRP2 reactive T cells (*Figure 2—figure supplement 1*). Indeed, overnight incubation of these cell lines with gp100-reactive T cells indicated they are equally susceptible to killing as the parental cell line (*Figure 2B*). To further corroborate that, we established cell lines from relapsed tumors following treatment with gp100-reactive T cells and assessed their responsiveness to treatment with gp100-reactive T cells. Indeed, these cell lines responded similarly to the parental cells to this treatment (*Figure 2C*). Next, we tested if tumor cell lines established following killing with TRP2- and gp100-reactive T cells would be less sensitive to killing by the same reactive T cells. First, we established melanoma cell lines that survived killing by either gp100 or TRP2-reactive T cells. These lines were then incubated with gp100 and TRP2-reactive T cells, and their rates of killing were compared to that of the parental cell line (illustrated in *Figure 2D*). We found that both B16F10 parental cells and the cell lines established following killing by reactive T cells were equally susceptible to T cell killing, (*Figure 2E*). Taken jointly, these results suggest that transient in vivo mechanisms rather than inherited ones govern resistance of relapsed tumors. In support of this hypothesis, RNAseq analysis indicated that all the established cell lines cluster within the same principal components, whereas the expression profile of freshly sorted tumor cells from immunotherapy-treated mice is markedly distinct (*Figure 2F*).

To better characterize the tumor cells that survive in mice following immunotherapy, we enzymatically digested treated tumors and sorted the live melanoma cells. Confocal analyses indicated that most of the tumor cells organize in clusters of several nuclei surrounded by a single membrane and cortical actin (*Figure 2G*). Transmission electron microscopy (TEM) of freshly sorted cells further

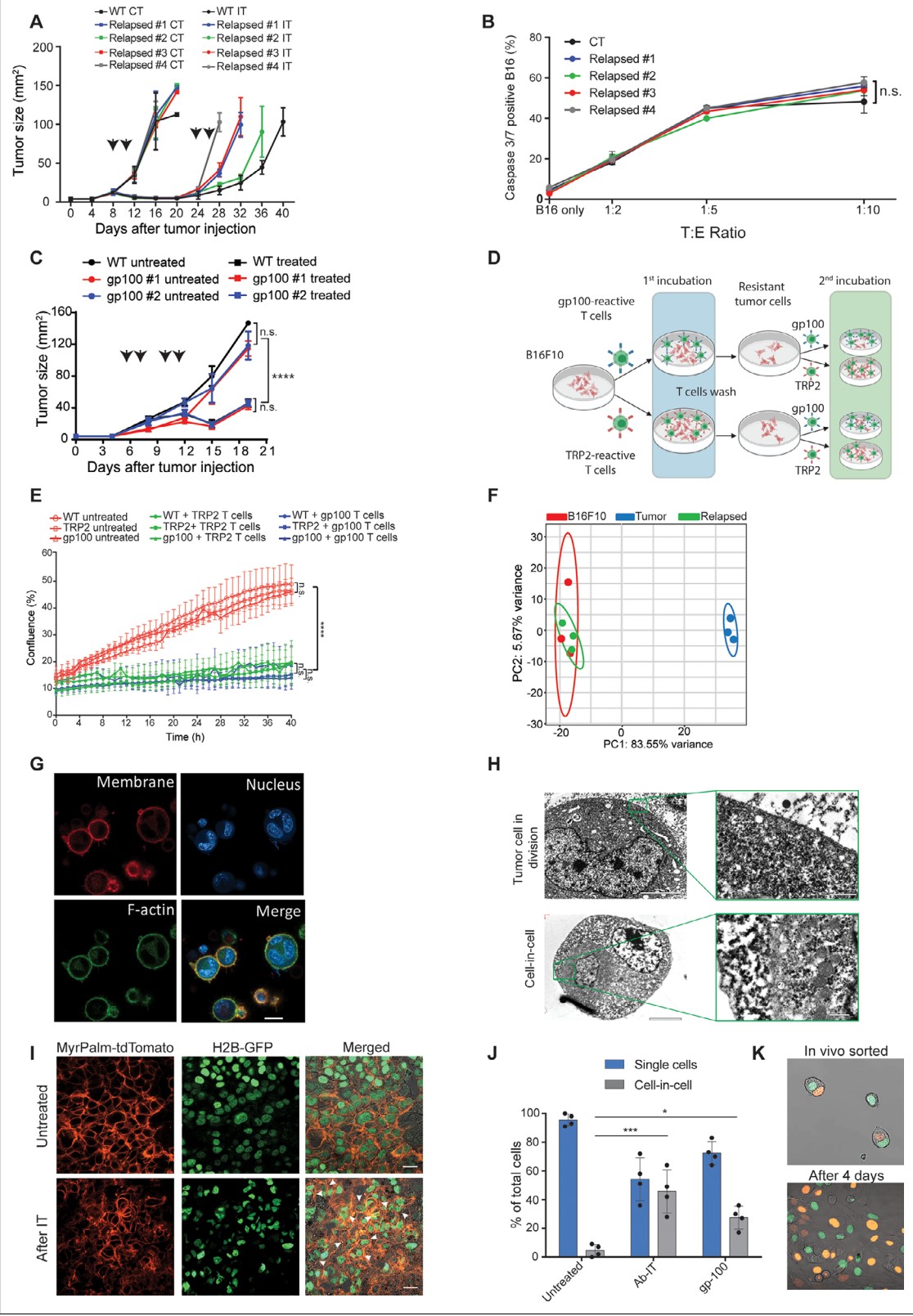

**Figure 2.** Tumor cells surviving immunotherapy organize in a transient cell-in-cell formation. (**A**) Tumor size (in mm²) of B16F10 control (WT) and cell-lines established from relapsed B16F10 tumors following re-treatments with DC adjuvant and anti-TRP1 tumor-binding antibodies (n=5). (**B**) Mean percentages of apoptotic tumor cells after incubation overnight with gp100-reactive CD8⁺ T cells (n=4). (**C**) Tumor size (in mm²) of B16F10 control (WT) and cell-lines established from relapsed B16 tumors following additional treatment with gp100-reactive T cell (n=4). Arrowheads indicate treatment.

*Figure 2 continued on next page*

*Figure 2 continued*

(**D**) Illustration of experimental design. (**E**) Confluence percentage of B16F10 following one or two incubations with gp100-reactive or TRP2-reactive CD8[+] T cells. (**F**) Principal Component Analysis (PCA) of B16F10 tumor cell lines freshly isolated from treated mice. (**G**) Representative images of B16F10 tumor cells sorted from tumors relapsed after immunotherapy. (**H**) Transmitting electron microscopy images of immunotherapy-relapsed B16F10 tumor cells. (**I**) Representative histological sections of untreated B16F10 tumors, and 5 days following immunotherapy. Arrowheads indicate cell-in-cell formation. (**J**) Mean percentage of cell-in-cell and single cells in B16F10 tumor-bearing mice, left untreated, treated with DC adjuvant and anti-TRP1 (Ab-IT), and treated with gp100-reactive T cells (gp100 ACT) (n=4). (**K**) Representative images of B16F10 labeled with H2B-tdTomato and H2B-GFP immediately after isolation from tumor-bearing mice treated with DC adjuvant and anti-TRP1, and after 4 days in culture. Experiments were repeated independently at least three times. Statistical significance was calculated using ANOVA with Tukey's correction for multiple comparisons (* denotes $p<0.05$, *** denotes $p<0.001$, and **** denotes $p<0.0001$). Error bars represent standard error. Scale bars = 20 μm (**G, I**), 5 μm (H left), 500 nm (H right).

The online version of this article includes the following video and figure supplement(s) for figure 2:

**Figure supplement 1.** The immunogenicity of tumor cell that relapsed following immunotherapy does not inherently alter, compared to their parental cells.

**Figure 2—video 1.** Spontaneous induction of cell-in-cell formation following sorting of single cells from tumors treated with immunotherapy, related to *Figure 2*.

https://elifesciences.org/articles/80315/figures#fig2video1

**Figure 2—video 2.** Spontaneous dissemination of cell-in-cell sorted directly from tumors following treatment with immunotherapy, related to *Figure 2*.

https://elifesciences.org/articles/80315/figures#fig2video2

corroborated a distinct segregation of the membranes and cytosols of the two cells. The inner cell was dense and seemed to have been compacted within another cell (*Figure 2H*). To ensure that these structures do not result from the isolation procedure, we also analysed histological sections of tumors, whose nuclei and cell membranes are fluorescently labeled. Indeed, immunotherapy-treated tumors showed increase prevalence of cell structures where one or more nuclei were surrounded by multiple membranes, especially in sites associated with tumor cell death (*Figure 2I*, *Figure 2—figure supplement 1C*). ImageStream FACS analysis as well as double blind count under confocal microscopy by pathologists indicated that about half the cells that survived immunotherapy are organized in such structures where multiple membrane layers surround the cell nuclei (*Figure 2J*).

To test if this formation was merely the result of incomplete cell division, we injected mice with tumor cells whose nuclei are labeled with GFP or tdTomato, and treated the mice with immunotherapy. About one third of the tumor cells were a mixture of both GFP and tdTomato nuclei (*Figure 2K*). Furthermore, monitoring the dynamics of single cells sorted from treated mice indicated that they spontaneously clustered in a cell-in-cell formation during their in vitro culture (*Figure 2—video 1*). Over time, all cells disseminated into single cells bearing morphological features similar to those of the parental cell line with no observed cells concomitantly expressing the two fluorophores (*Figure 2—video 2*).

## Tumor-reactive T cells induce transient cell-in-cell structure

To test whether a specific immune cell type mediates these cell-in-cell formations, we isolated the main effector cells from tumor-bearing mice and incubated them overnight with B16F10 tumor cells. We found that only T cells—mainly CD8[+], and with slower kinetics also CD4[+]—induce a cell-in-cell formation with the same special characteristics as the ones observed in vivo following immunotherapy (*Figure 3A, B*, *Figure 3—figure supplement 1*, *Figure 3—video 1*). Consistently, no cell-in-cell structures were observed in Nude SCID- gamma[-/-] (NSG) mice treated with immunotherapy (*Figure 3C*).

We next tested whether this phenomenon occurs in other tumor cell types. Indeed, the same structures were also observed in 4T1 mammary carcinoma incubated with allogeneic T cells, but not in immortalized mammary epithelial cells (*Figure 3—figure supplement 1*). In contrast, immortalized mammary epithelial cells were completely killed by allogeneic T cells and did not form cell-in-cell structure (*Figure 3—figure supplement 1*) suggesting this phenomenon is restricted to tumor cells. We also tested whether tumor cells from different tissue origins can organize in cell-in-cell formation. To this end, we incubated tumor cell lines with allogeneic T cells and quantified the number of cell-in-cell structures after 24 hr. We found significant differences between tumor cell lines; on one end are pancreatic tumors and B cell lymphoma, which do not generate cell-in-cell structures, while on the other end, almost all the colon and ovarian carcinoma tumor cells that survive T cell killing, were organized in such structures (*Figure 3D*, *Figure 3—figure supplement 1E–H*).

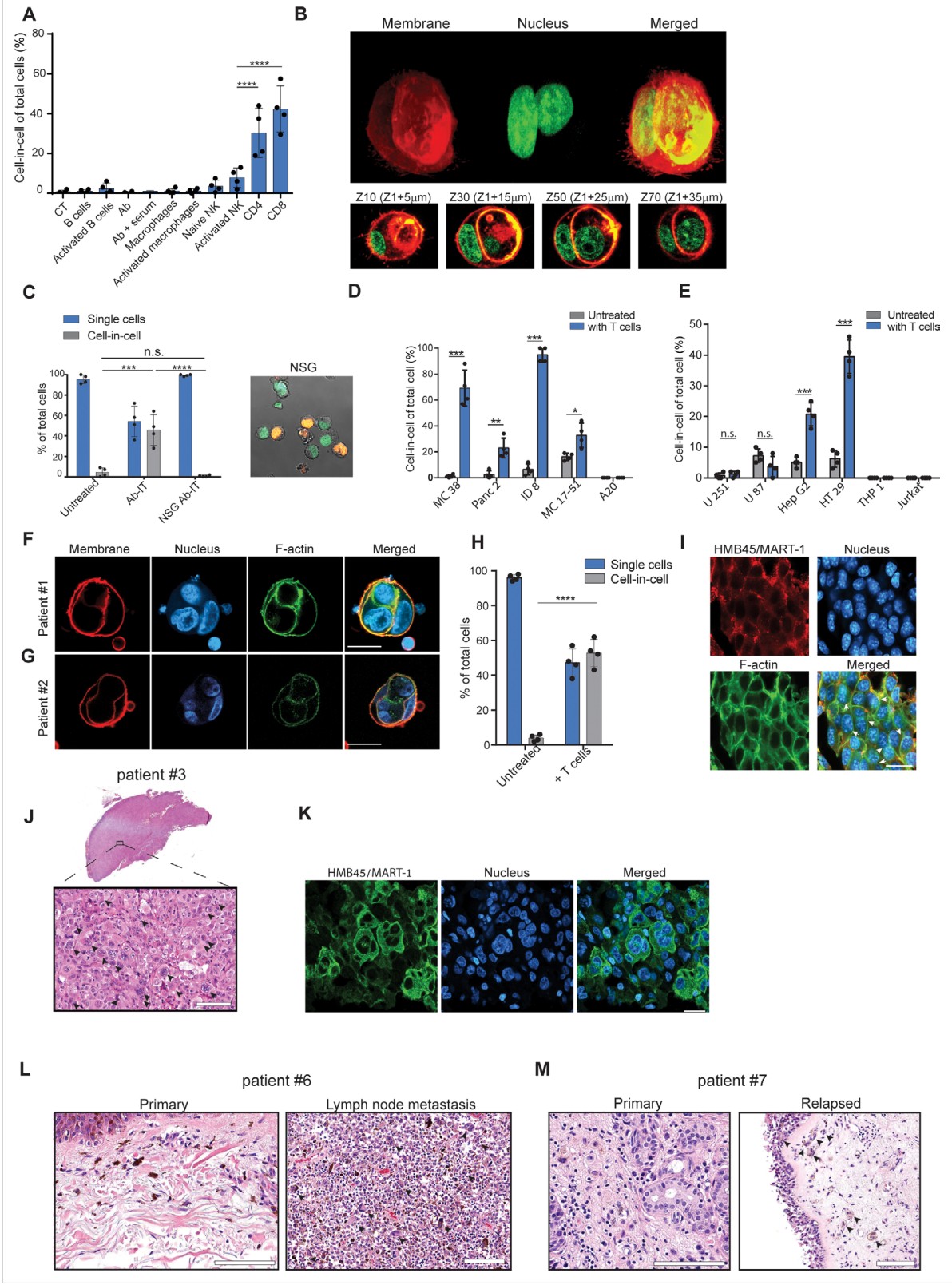

**Figure 3.** Tumor-reactive T cells induce transient cell-in-cell structure. (**A**) Mean percentage of cell-in-cell tumor formations following incubation of B16F10 cells with immune cells (n=4). (**B**) Representative 3D projection and horizontal sections (Z-stack) of B16F10 cells incubated with gp100-reactive CD8[+] T cells. (**C**) Mean percentage of cell-in-cell tumor formations and single cells in tumor-bearing NSG[-/-] mice treated with DC adjuvant and anti-TRP1 antibodies (n=4). (**D**) Mean counts of cell-in-cell tumor formations in mouse cancer cell lines cultured overnight with activated CD8[+]

*Figure 3 continued on next page*

Figure 3 continued

T cells (n=4). (**E**) Mean counts of cell-in-cell tumor formations in human cancer cell lines cultured overnight with activated allogeneic CD8[+] T cells (n=4). (**F–G**) Representative images of freshly-isolated human melanoma cells incubated overnight with autologous tumor-reactive T cells. (**H**) Mean percentage of cell-in-cell tumor formations following incubation of tumor cells isolated from melanoma patient with autologous tumor-reactive T cells (four technical replicates). (**I**) Histological section of human melanoma in NSG[-/-] mice 7 days after treatment with patient autologous TIL and high-dose IL-2. (**J**) H&E staining of histological sections of involved lymph node from metastatic melanoma patient. (**K**) Immunostaining of involved lymph nodes from metastatic melanoma patient. (**L**) H&E staining of histological sections of primary and sentinel lymph nodes from untreated stage IV melanoma patients. (**M**) H&E staining of histological sections of untreated primary and relapsed melanoma patient. Statistical significance was calculated using ANOVA with Tukey's correction for multiple comparisons (* denotes p<0.05, ** denotes p<0.01 *** denotes p<0.001, and **** denotes p<0.0001). Error bars represent standard error. Scale bars = 20 μm.

The online version of this article includes the following video and figure supplement(s) for figure 3:

**Figure supplement 1.** Murine tumor cell lines undergo cell-in-cell formation incubation with reactive T cells, Related to *Figure 2* and *Figure 3*.

**Figure supplement 2.** Human tumor cells undergo cell-in-cell formation following incubation with reactive T cells, Related to *Figure 2* and *Figure 3*.

**Figure 3—video 1.** B16 cell-in-cell induction in vitro following incubation with gp100-reactive T cells, related to *Figure 3*.
https://elifesciences.org/articles/80315/figures#fig3video1

**Figure 3—video 2.** HT29 cell-in-cell induction in vitro following incubation with allogeneic T cells, related to *Figure 3*.
https://elifesciences.org/articles/80315/figures#fig3video2

We then tested whether similar structures are also formed by human cancer. To this end, we incubated tumor cell lines with pre-activated allogeneic T cells from healthy donors. Similar to our finding in mouse cells, the vast majority of breast, colon and melanoma tumors that survived killing had organized in transient cell-in-cell structures (*Figure 3E* and *Figure 3—figure supplement 2*). Moreover, a 3-day recording of the interactions between colorectal adenocarcinoma and activated T cells highlighted the dynamic nature of these cell-in-cell structures, as tumor cells constantly form and disseminate from it. These results further stress that most single cells will undergo cell death (*Figure 3—video 2*). In contrast, T cell and myeloid lymphomas as well as glioblastomas did not form such structures (*Figure 3E* and *Figure 3—figure supplement 2A–G*). We also assessed that in a more clinically relevant model. To this end, we isolated melanoma cells and their autologous reactive T infiltrating lymphocytes (TIL) from freshly resected primary melanoma tumors. Overnight incubation of patient-derived melanoma and their corresponding reactive TIL, resulted in vast tumor cell killing. Approximately half of the tumor cells that survived in culture were organized in a cell-in-cell formation (*Figure 3F–H*). To further corroborate these observations, melanoma cells from human patients were also injected into NSG[-/-] mice and tumors were allowed to grow. Once tumors reached approximately 40 mm$^2$, mice were injected i.v. with $1 \times 10^7$ autologous TIL and with high-dose IL-2. Tumor sizes regressed drastically for several days until reaching a minimum size of 9 mm$^2$, but regrew afterwards. Histological sections of tumors at this point indicated high prevalence of tumor cells organized in cell-in-cell formation (*Figure 3I*). We also analysed histological sections of multiple organs from four stage IV melanoma patients undergoing surgical resection of their primary, and metastatic draining lymph nodes. In all patients, cell-in-cell tumor formations were highly abundant in the T cell zone of the draining lymph nodes, but not in the primary tumors (*Figure 3J–L* and *Figure 3—figure supplement 2H*). In another patient with untreated recurrent melanoma, we observed that the vast majority of cells in the primary tumors were single cells, while recurrent tumors were highly enriched with cell-in-cell formations (*Figure 3M*).

## IFNγ induces membrane-bound proteins on T cells that mediate tumor cell-in-cell formations

To elucidate the process by which T cells induce a cell-in-cell formation, we initially compared their interactions with tumor cells to that of other immune cells using high-resolution SEM. Interestingly, tumor-reactive T cells, but not PMA plus ionomycin-activated NK cells or LPS-activated macrophages, express the intact granules on their cell membrane (*Figure 4A*, *Figure 4—figure supplement 1*). To test if these T cell-secreted granules can mediate cell-in-cell structures, we incubated splenic T cells with immobilized anti-CD3 and anti-CD28 and isolated intact granules on iodixanol gradient by ultra-centrifugation (*Casey et al., 2007*). Secreted granules were washed and added to a culture of tumor cells. Similar to activated T cells, secreted granules were sufficient to induce cell-in-cell formation.

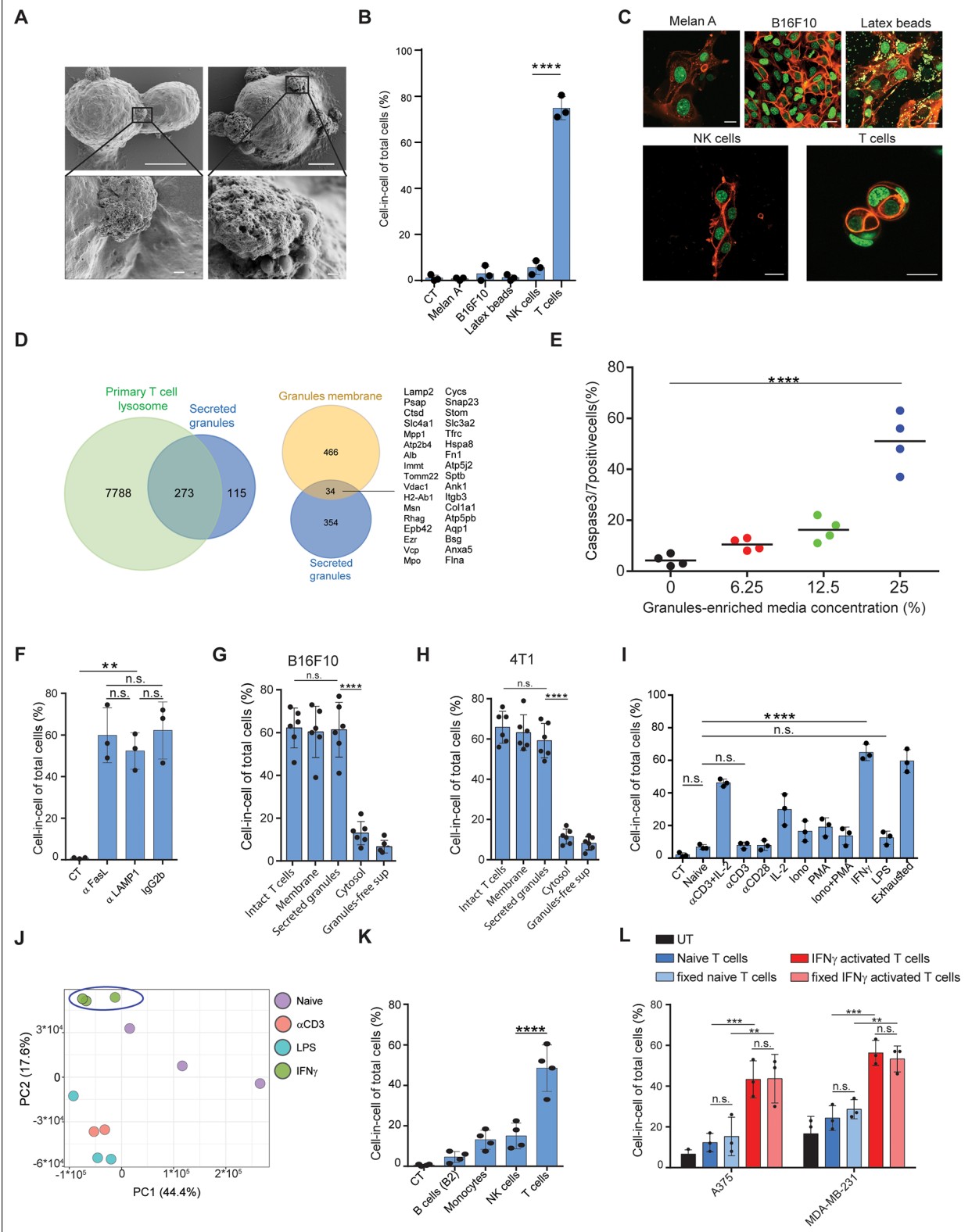

**Figure 4.** IFNγ induces membrane-associated proteins on T cells that mediate tumor cell-in-cell formation. (**A**) SEM analysis of B16F10 incubated overnight with gp100-reactive CD8+ T cells. (**B**) Mean percentage of cell-in-cell formation of B16F10 incubated overnight with latex beads, B16F10 and Melan A melanosomes, NK- and T cell-derived granules (n=3). (**C**) Representative images of B16F10 incubated overnight with latex beads, B16F10 and Melan A melanosomes, NK- and T cell-derived granules. (**D**) Overlap of the total genes, detected by mass-spectrometry of T cell secreted granules and

*Figure 4 continued on next page*

*Figure 4 continued*

genes related to primary human T cell granules (left diagram) or granules-membrane related genes (right diagram). (**E**) Mean percentages of apoptotic tumor cells following overnight incubation with T cell-derived granules. (**F**) Mean percentage of cell-in-cell formation following overnight incubation with granules-blocking antibodies (n=3). (**G**) Mean percentage of B16F10 cell-in-cell formations following overnight incubation with T cells compartments (n=6). (**H**) Mean percentage of 4T1 cell-in-cell formations following overnight incubation with T cells compartments (n=6). (**I**) Mean percentage of cell-in-cell formation following incubation with T cells pre-activated with different stimulators (n=3). (**J**) PCA of expression profiles of naïve and activated CD8$^+$ T cells (**K**) Mean percentage of cell-in-cell formations following overnight incubation with IFNγ-activated immune cells. (n=4). (**L**) Mean percentage of human cell-in-cell formations following overnight incubation with IFNγ-activated CD8$^+$ T cells from healthy donors. (n=3). Experiments were repeated independently at least three times. Statistical significance was calculated using ANOVA with Tukey's correction for multiple comparisons (**denotes $p<0.001$, *** denotes $p<0.0001$, **** denotes $p<0.0001$). Error bars represent standard. Scale bars = 5 μm (A top), 500 nm (A bottom), 20 μm (**C**).

The online version of this article includes the following figure supplement(s) for figure 4:

**Figure supplement 1.** IFNg-activated T cells induce cell-in-cell tumor formation, Related to *Figure 4*.

Next, we tested if this formation can be induced by extracellular vesicles secreted by other cells. Hence, B16F10 cells were cultured overnight with extracellular vesicles from activated NK, macrophages, immortalized melanocytes (Melan A), B16F10-derived granules, or commercial 1 μM latex beads. However, none of these extracellular vesicles induced a cell-in-cell formation, suggesting this it is mediated by molecules that are highly enriched in activated T cells (*Figure 4B, C*).

To shed light on the nature of these granules, we initially analysed them under TEM. The vast majority of isolated granules had a dark center due to high absorbance of uranyl acetate, rang at about 500 nm in diameter, and their luminal matrix was bounded by a bilayer membrane (*Figure 4— figure supplement 1*). Similar characteristics were also observed in granules on the membranes of activated CD8$^+$ T cells incubated with tumor cells (*Figure 4—figure supplement 1*). We next analysed the protein content of the purified granules by mass spectrometry. Overall, we identified approximately 400 proteins and out of those 273 were shared primary T cell secreted granules. (*Figure 4D*). Consistently, incubation of these granules with tumor cells induced significant cell death (*Figure 4E*, *Figure 4—figure supplement 1*). Whether these granules are indeed lysosomes, or other intact lytic granules, such as super macromolecules attack particles (*Bálint et al., 2020*), remain beyond the scope of this work. Along these lines, whether these granules are released from dying T cells or through the active mechanism of exocytosis currently remains unknown. Most importantly, however, incubation of tumor cells with gp100-reactive T cells in the presence of antibodies that block the interactions with lysosomes/secreted granules (i.e. LAMP1, CD95, or CD63) did not alter T cells capacity to induce a cell-in-cell formation, suggesting that these molecules are also expressed on T cell membranes (*Figure 4F*). Indeed, incubation of tumor cells with extraction of membrane-associated proteins, but not cytosolic or secreted proteins from activated T cells, induce tumor cell-in-cell formation (*Figure 4G–H*). To elucidate the type of activations that predominantly induce these molecules, we incubated naïve splenic T cells overnight with various stimulators, washed them extensively, and incubated them with tumor cells. We found that T cells activated with IFNγ, and to a lesser extent anti-CD3 plus high-dose IL-2, induce the most cell-in-cell formation (*Figure 4I*, *Figure 4—figure supplement 1*). RNAseq analyses indicated that the expression profile of IFNγ activated T cells differ markedly than that induced by anti-CD3 or LPS, which do not induce cell-in-cell formation in tumor cells (*Figure 4J*, *Figure 4—figure supplement 1*). We next tested if other IFNγ -activated cells can induce a cell-in-cell formation, or whether it is restricted to activated T cells. We found that IFNγ -stimulated NK cells and macrophages, but not B cells, could induce a tumor cell-in-cell formation, albeit at a lower percentage compared to activated T cells (*Figure 4K*). Similarly, human T cells from healthy donors that were pre-activated with IFNγ efficiently induced cell-in-cell formation in allogeneic tumor cells (*Figure 4L* and *Figure 4—figure supplement 1*). Overall, these results suggest that proteins that are predominantly, but not exclusively, expressed on both the membrane of IFNγ-activated T cells and on secreted granules mediate cell-in-cell formation.

## Cell-in-cell formation spatially prevents the penetration of T cell-derived lytic granules to the inner tumor cells

Next, we assessed if cell-in-cell formation protects tumor cells from T cell-mediated killing. To this end, we compared the viability and apoptosis rates of single tumor cells to that of cell-in-cell. Initially, cell-in-cell formation was induced by overnight incubation of melanoma cells with T cells-secreted

granules. Next, we added different ratios of gp100-reactive CD8$^+$ T cells and measured caspase 3/7 activity in the tumor cells after 6 hr and overnight. In all assays, we found that single tumor cells were significantly more susceptible to killing by reactive T cells, compared to that of cell-in-cell tumors (*Figure 5A*). Pre-incubation of tumor cells with granules isolated from activated B cells and macrophages (which do not induce cell-in-cell formation) provided no protection to tumor cells from gp100-reactive T cells (*Figure 5—figure supplement 1*). We next tested if cell-in-cell formation also endows protection from immunotherapy in vivo. To this end, cell-in-cell tumors were induced in vitro, before injection to animals, by overnight incubation with T cell secreted granules, or in vivo by injection of T cell-derived secreted granules. Cell-in-cell tumors that were induced in vitro have grown slower compared to single tumor cells, but experienced only minor regression following administration of immunotherapy. Similarly, cell-in-cell formation induced in vivo showed only benign response to immunotherapy. In sharp contrast, single melanoma cells were almost completely abrogated following therapy (*Figure 5B*).

To assess the type of protection cell-in-cell formation provides to tumor cells, we examined their interactions with T cells under scanning electron microscope (SEM). We found that multiple T cells were attached to cell-in-cell tumors and many large pores were observed on the outer cell membrane (*Figure 5C*). These pores were almost completely absent upon co-culture with reactive CD8$^{-/-}$ T cells from perforin$^{-/-}$ mice (*Figure 5D*), which have impaired tumor cell killing abilities (*Figure 5—figure supplement 1*). Furthermore, measuring the length and the duration of the immunological synapse under super-resolution microscopy showed that no reduction compared to single tumor cells (*Figure 5E–G*, *Figure 5—figure supplement 1*). Taken jointly, these results strongly suggest that cell-in-cell structures do not evade killing by escaping CD8$^+$ T cell recognition. Alternatively, we considered the possibility that lytic enzymes derived from T cell do not penetrate these cellular structures. Immunostaining of T cells attacking cell-in-cell tumors, indicated that the distribution of granzyme B and perforin is almost limited to the outer cell (*Figure 5H–I*, *Figure 5—figure supplement 1*). While these interactions were sufficient to kill single tumor cells, they were insufficient to induce caspase 3/7 activity in the inner cell, leaving it intact and alive (*Figure 5G–J*, and *Figure 5—video 1*).

Lastly, we compared the sensitivity of single-tumor cells to that of cell-in-cell tumors to additional antibiotics including free radicals and chemotherapies. Similar to our results with T cells, in all the tested compounds, cell-in-cell formation tumors survived longer and at higher concentrations compared to their parental cell lines (*Figure 5K* and *Figure 5—figure supplement 1*).

## T-cell-mediated cell-in-cell formation is governed by STAT3 and EGR1 signaling

We then set out to understand what signaling cascade governs cell-in-cell structures. Initially, we sought to test if the observed results were cell fusion. Thus, we incubated tumor cells whose cytosols were labeled with either Wasabi or tdTomato with tumor-reactive T cells. Confocal analysis indicated that each cell type in the cell-in-cell formation maintained its cytoplasm, and no mix between colors was detected (*Figure 6—figure supplement 1*). Furthermore, long-term follow-up of cell dissemination from this structure indicated that each cell maintains its initial single labelling color (*Figure 6—video 1* and *Figure 6—figure supplement 1*). To corroborate this, we also incubated tumor cells whose membrane, nucleus, and F-actin were labeled with different fluorophores. Similarly, each cell in this formation was separated and maintained the integrity of its original cell components (*Figure 6A*). Given the similarity of this formation to entosis, we used ROCK inhibitor, which is the key regulator of this process. Indeed, blocking ROCK almost completely prevented T cell-mediated cell-in-cell formation (*Figure 6B, C*). We then tested whether the molecular machinery reported to mediate tumor spontaneous entosis applies to govern the current cell-in-cell structure. However, we observed no increase or changes in the cellular localization of phosphorylated β catenin, E-cadherin, and phosphorylated integrin β1 (26) (*Figure 6—figure supplement 1*) suggesting other mediators promote this entosis. Furthermore, there was no reduction in cell-in-cell formation upon blocking of E- and N- cadherins, or inhibition of Wnt signaling (*Figure 6D–E*). In sharp contrast, disruption of actin filaments, blocking of mRNA synthesis or protein production completely abrogated tumor cells' capacity to form a cell-in-cell formation, suggesting that the structure requires de novo synthesis of genes (*Figure 6D–E*).

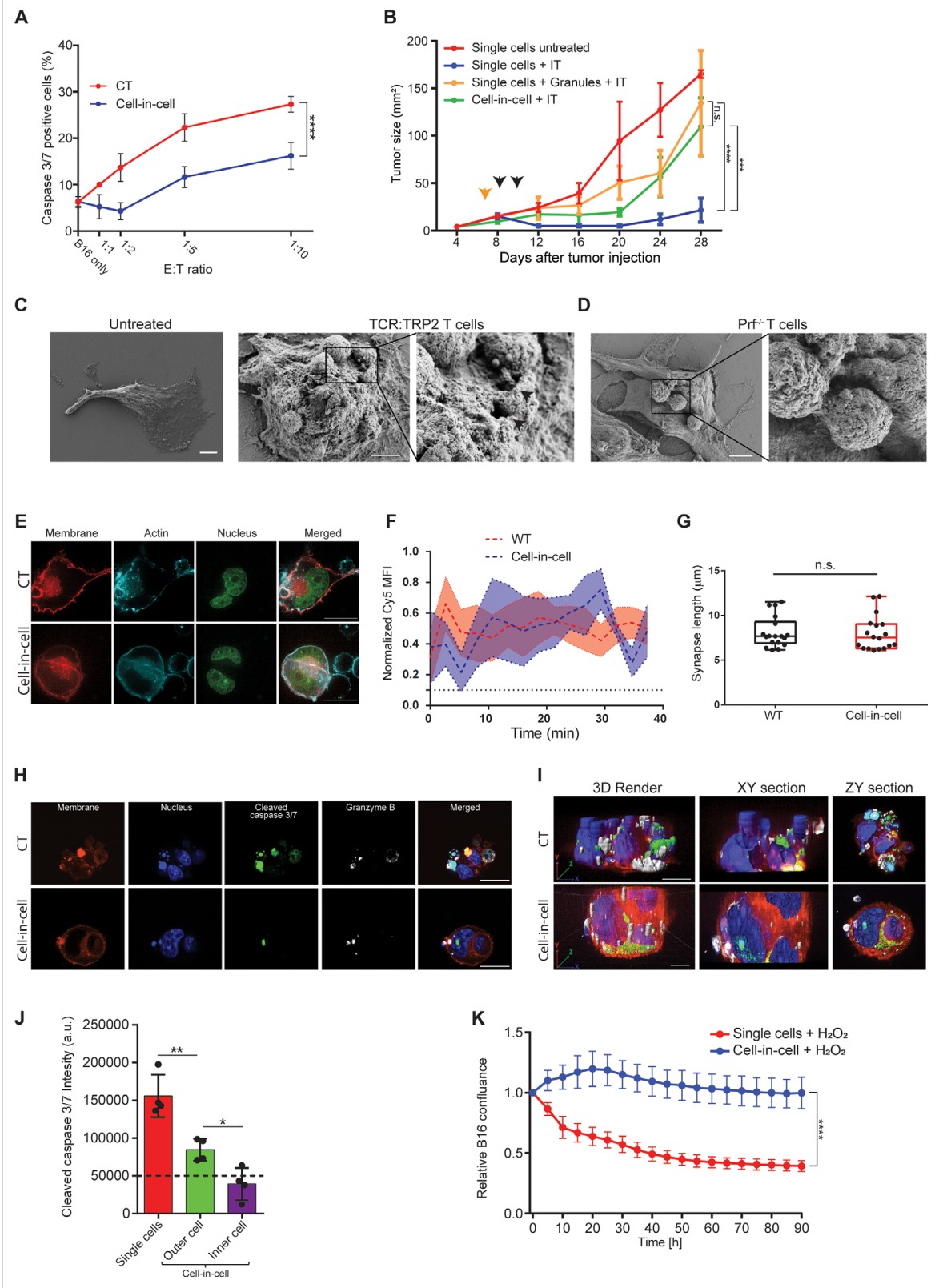

**Figure 5.** Cell-in-cell formations spatially prevent the penetration of T cell-derived lytic granules to the inner tumor cells. (**A**) Mean percentages of apoptotic tumor cells following overnight incubation with gp100-reactive CD8[+] T cells. (**B**) B16F10 tumor size following treatment with DC adjuvant and anti-TRP1 antibodies (IT). B16F10 were injected as single cells, with or without injection of T cell-secreted granules (Granules), or as cell-in-cell. Black arrowheads indicate IT, yellow arrowhead indicates granules injection. (**C–D**) SEM analysis of B16F10 incubated with TRP2-reactive CD8[+] T cells

*Figure 5 continued on next page*

*Figure 5 continued*

from WT (**C**) or Prf1[-/-] mice (**D**). (**E**) Confocal plane image of B16F10 incubated with tumor-reactive CD8[+] T cells. (**F**) Normalized intensity of T cell's cortical actin (Cy5) over time at immunological synapses (n=5). (**G**) Immunological synapse length between B16F10 and tumor-reactive CD8[+] T cells (n=18). (**H**) Confocal plane image of B16F10 incubated with tumor-reactive CD8[+] T cells. (**I**) 3D rendering and surface detection and 3D sections (XY and ZY planes) of B16F10 cells incubated with tumor-reactive CD8[+] T cells. (**J**) Mean fluorescence intensity of cleaved caspase 3/7 in B16F10 tumor cells following incubation with gp100-reactive CD8[+] T cells (n=3). (**K**) Relative confluence over time of B16F10 single cell or cell-in-cell, cultured following incubation with $H_2O_2$ (n=3). Experiments were repeated independently at least three times. Statistical significance was calculated using ANOVA with Tukey's correction for multiple comparisons (* denotes p<0.05, ** denotes p<0.01, *** denotes p<0.001, **** denotes p<0.0001). Error bars represent standard error. Scale bars = 5 µm (**C, D**), 20 µm (**E, H, I**).

The online version of this article includes the following video and figure supplement(s) for figure 5:

**Figure supplement 1.** Cell-in-cell tumor formations are recognized but not killed by reactive T cells, related to *Figure 5*.

**Figure 5—video 1.** B16 cell-in-cell interaction with gp100-reactive T cells, related to *Figure 5*.

https://elifesciences.org/articles/80315/figures#fig5video1

To assess what genes govern this formation, we compared the gene signature of untreated B16F10 cells, to cell-in cell formation induced following incubation with T-cell-derived secreted granules and to that of tumor cells sorted from in vivo five days after immunotherapy. Over 400 genes were increased in cell-in-cell formation induced by T cell secreted granules compared to untreated B16F10 cells, 215 of which were also upregulated by the tumor cells sorted from treated animals (*Figure 6F*, ). KEGG analysis further indicated that multiple signaling cascades, including the JAK/STAT3 axis and FGF-receptors downstream pathways, are enriched in cell-in-cell formation generated in vivo, with approximately 80 significantly elevated genes relating to these pathways (*Figure 6—figure supplement 1*, ). Indeed, EGR1 and STAT3 expression was significantly elevated in cell-in-cell tumors (*Figure 6G*). Both mice and human tumors incubated with reactive T cells had elevated their p-STAT3 levels, in comparison to untreated tumor cells (*Figure 6—figure supplement 1*), suggesting this mechanism is conserved across species. Since these results may also reflect adaptation to the tumor microenvironments, we next corroborate the necessity of these factors to cell-in-cell formation ex vivo. Overexpression of either STAT3 or EGR1 was sufficient to induce cell-in-cell formation without additional stimulation (*Figure 6H*). Furthermore, these cells were significantly more resistant to killing by CD8[+] T cell, compared to sham transfected cells (*Figure 6I*). We also tested if inhibiting these genes would reduce cell-in-cell tumor formation. Inhibition of STAT3 and EGR1, but not MAPK3 and EGR2, significantly abrogated the capacity of tumor cells to form cell-in-cell structures upon incubation with IFNγ-stimulated T cells and with T cell secreted granules (*Figure 6J* and *Figure 6—figure supplement 1*). In order to integrate cell-in-cell inhibition to an in vivo therapy, we first tested the effect of small molecule inhibitors on tumor cell formation. We found that inhibition of STAT3, or EGR pathway, completely prevented cell-in-cell formation upon incubation with reactive T cells or T cell-secreted granules (*Figure 6—figure supplement 1*). Since blocking EGR1 also reduced T cell viability, we then set out to establish a treatment protocol that combines STAT3 inhibition (which also inhibits T cell activity) and immunotherapy. To this end, mice bearing palpable tumors were injected with Stattic or with PBS for two consecutive days. On the second day, we sub-lethally irradiated the mice and injected them with $5 \times 10^6$ gp100-reactive T cells and IL-2. In another model, we treated mice for two days with Stattic, followed by treatments with anti-CD40, TNFα, and anti-TRP1. Recurrent tumors were treated with the same regimen. In both models, injection of Stattic alone had no effect on tumor growth. We found, however, that injection of Stattic prior to administration of immunotherapy partially restored the responsiveness of tumors that re-occur following immunotherapies (*Figure 6K–L*). While tumor cell sensitization may be a result from multiple mechanisms, these results stress the benefit of combining immunotherapy with STAT3 inhibition.

## Discussion

Cancer immunoediting has been the primary paradigm for explaining how tumors escape T-cell immunity and the framework for most recent immunotherapies (*Dunn et al., 2002*; *Mittal et al., 2014*). In support of this view, fibrosarcoma clones emerged in T-cell-deficient mice and were deleted upon injection to immunocompetent mice (*Matsushita et al., 2012*; *DuPage et al., 2012*). Immunoediting is further supported by clinical observations in which tumors relapse by omitting their expressed

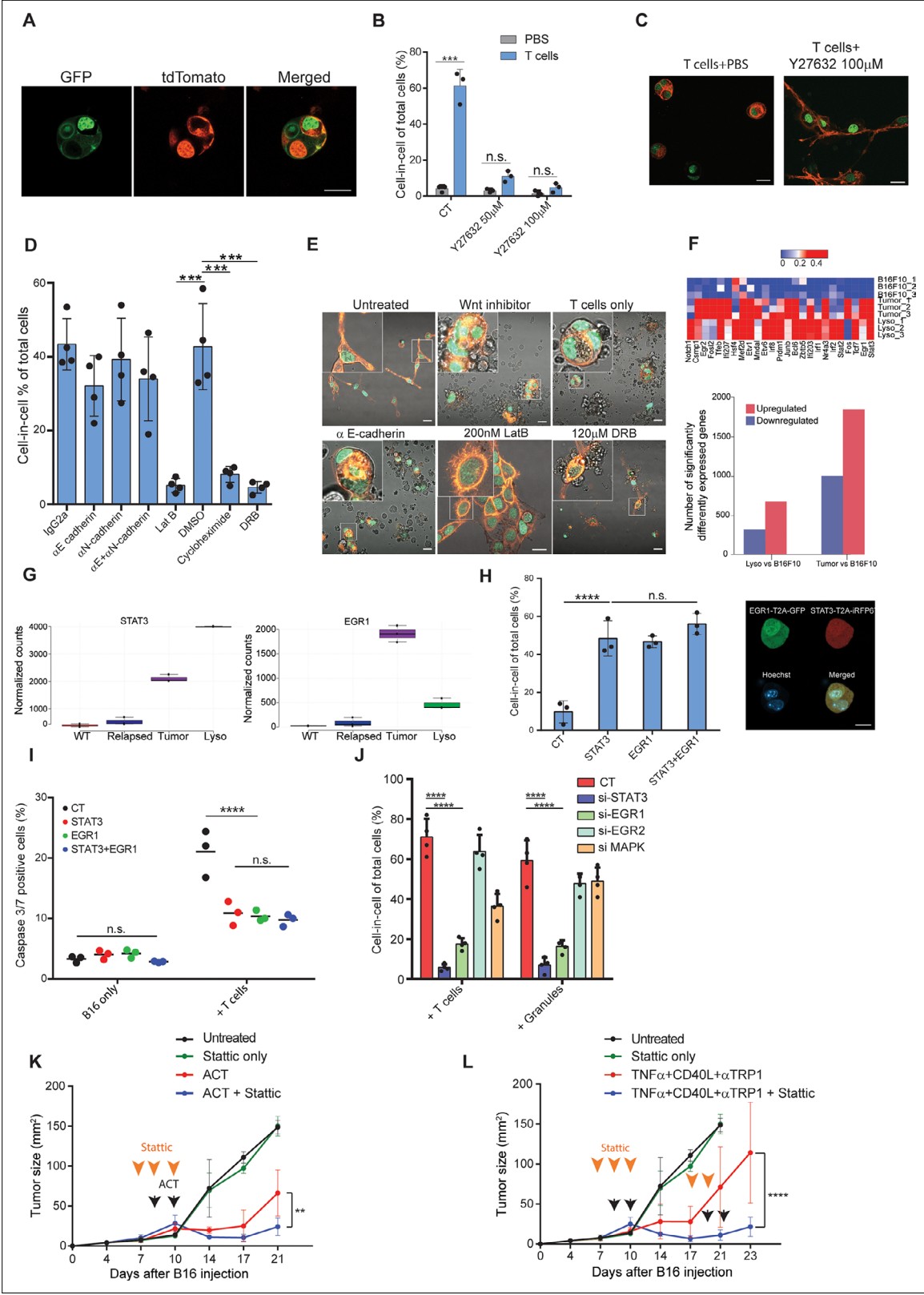

**Figure 6.** STAT3 and EGR1 signaling govern T cell-mediated cell-in-cell tumor formation. (**A**) Representative images of B16F10, co-expressing Lifeact-GFP and H2b-tdTomato or MyrPalm-tdTomato and H2b-GFP, following incubation with gp100-reactive T cells. (**B**) Mean percentage of cell-in-cell tumor formations in B16F10 following overnight incubation with gp100-reactive CD8⁺ T cells with or without ROCK inhibitor (n=3). (**C**) Representative images of cell-in-cell tumor formations in B16F10 following overnight incubation with gp100-reactive CD8⁺ T cells with or without ROCK inhibitor. (**D**) Mean

*Figure 6 continued on next page*

*Figure 6 continued*

percentage of cell-in-cell tumor formations in B16F10 cells following overnight incubation with specific inhibitors and reactive CD8$^+$ T cells (n=4). (**E**) Representative images of B16F10 cells treated with inhibitors and incubated overnight with gp100-reactive CD8$^+$ T cells. (**F**) Significantly increased genes in B16F10 cells incubated with T cell-derived granules (Lyso) or isolated directly from relapsed tumors (Tumor), compared to B16F10 control cells (WT) (Bottom) and relative expression of the top 25 genes (Top) (n=3). (**G**) STAT3 and EGR1 expression levels in B16F10 cells isolated directly from relapsed tumors (Tumor) and after incubation with T-cell-derived granules (Lyso) compared to B16F10 control cells (WT) (n=3) (**H**) Mean percentage and representative images of cell-in-cell tumor formations in B16F10 48 hours after transfection with STAT3-T2A-iRFP670, EGR1-T2A-GFP or both (n=3). (**I**) Mean percentage of apoptotic B16F10, transfected with STAT3-T2A-iRFP670, EGR1-T2A-GFP or both, following incubation with tumor reactive T cells (n=3). (**J**) Mean percentage of cell-in-cell tumor formations in B16F10, transfected with siRNA, following incubation with tumor reactive T cells or T cells secreted granules. (**K–L**) B16F10 tumor size in mice treated with gp100-reactive T cells (ACT) (**K**) or Dc adjuvant and anti-TRP1 antibodies (**L**) with or without Stattic (n=4). Orange arrowheads indicate Stattic treatments and black arrowheads indicate injection of immunotherapies. All experiments were repeated independently at least three times. Statistical significance was calculated using ANOVA with Tukey's correction for multiple comparisons (**denotes p<0.01, *** denotes p<0.001, **** denotes p<0.0001). Error bars represent standard error. Scale bars = 20 µm.

The online version of this article includes the following video, source data, and figure supplement(s) for figure 6:

**Source data 1.** Significantly elevated genes in both B16F10 cells incubated with T cell secreted granules and tumor cells sorted from treated animals, related to *Figure 6*.

**Source data 2.** STAT3 and EGR1 pathways regulate cell-in-cell tumor formation, related to *Figure 6*.

**Source data 3.** Log2 expression of genes related to cancer pathways in tumor cells sorted from treated animals compared to B16F10 WT, reated to *Figure 6*.

**Figure supplement 1.** STAT3 and EGR1 pathways regulate cell-in-cell tumor formation, related to *Figure 6*.

**Figure 6—video 1.** B16 cell-in-cell dissemination in vitro following T cell wash, related to *Figure 6*.
https://elifesciences.org/articles/80315/figures#fig6video1

antigen (*Verdegaal et al., 2016*; *von Boehmer et al., 2013*; *Tran et al., 2016*). While it is clear that immunogenic clones can be eliminated by T cells, the presence of immunogenic clones that express neoantigens, despite having infiltrated reactive T cells, has been well documented (*Straetemans et al., 2015*). Given the complexity of such a procedure, only a handful of clinical studies analysed changes in the tumor neoantigen landscape during immunotherapy. Zaretsky JM. et al. found that relapsed melanoma following treatment with pembrolizumab shares the majority of neoantigen with their corresponding primary tumors (*Zaretsky et al., 2016*). Consistently, Anagnostou V. et al. found that non-small cells carcinoma (NSCC) relapsed following treatment with anti-PD1 and CTLA-4 checkpoint blockade bear similar neoantigen load, with only a relatively small number of putative deleted clones (*Anagnostou et al., 2017*). In a recent study, Yost et al. demonstrated replenishment of T cell clones in basal cell carcinomas patients treated with pembrolizumab. Interestingly, three out of four analysed patients exhibited similar neoantigen burden (*Yost et al., 2019*). These studies therefore suggest that immunotherapy-resistant tumors are composite also of immunogenic clones, which can be recognized by the host T cell, and that non-genetic mechanisms underly their persistence.

Alternative explanation regarding how immunogenic clones can survive T-cell pressure is attributed to the capacity of relapsed tumors to promote a suppressive microenvironment (*Sharma et al., 2017*; *Topalian et al., 2015*). Indeed, over the past decade multiple parameters have been suggested to predict patient's response to immunotherapy (*Thorsson et al., 2018*; *Petitprez et al., 2020*). However, why tumors reoccur in patients undergoing complete response remains unknown. Here, we found that tumor cells escape T-cell-based immunotherapies by forming a transient cell-in-cell structure, and not by immunoediting. These structures generate multiple membrane layers, thus reducing the probability of compounds' internalization leading to increased tumor-cell survival. It should be taken under consideration that most of our results were achieved using immunotherapy of DC adjuvant in combination with tumor-binding antibodies. Albeit reproduced with adoptive cell transfer, whether cell-in-cell tumor-formations form as an escape mechanism for immunotherapy following checkpoint blockade is still not clear. Given the limited efficacy of classical checkpoint blockade therapy in animal models, it is challenging to employ it as a model for tumor relapse.

Many cell-in-cell formations have been described before including phagocytosis, cell cannibalism, and entosis (*Krishna and Overholtzer, 2016*). Recently, some of these structures have been shown to enable tumor-cell survival after chemotherapy treatment by engulfing and digesting neighboring cells (*Tonnessen-Murray et al., 2019*). Although the present cell-in-cell structural falls within the definition entosis, it differs in several key elements: First, it is triggered predominantly by reactive T cells, rather

than glucose and nutrient starvation (*Hamann et al., 2017*), mitosis (*Durgan et al., 2017*), or loss of adherence (*Overholtzer et al., 2007*). Additionally, it requires de novo synthesis of genes regulated by STAT3, and not through β-catenin signaling (*Overholtzer et al., 2007*). Most importantly, the structure described here is reversible and only rarely leads to cell cannibalism and apoptosis.

We found that only tumor cells that are capable to undergo 'spontaneous' entosis due to loss of cell adhesion, can form cell-in-cell structure following interactions with reactive T cells. Why hematopoietic cells do not undergo 'spontaneous' or T-cell-mediated entosis is an intriguing question that is not yet not clear. In general, hematopoietic cells do no undergo entosis but rather emperipolesis (*Krishna and Overholtzer, 2016*). It may be that since these cells are equipped with different sets of integrins compared to epithelial cells, they may lack the relevant integrins and cadherins that facilitate cell-in-cell tumor formation (*Gupta et al., 2017*). Instead, emperipolesis is thought to governed by hematopoietic-specific integrins, such as ICAM-1 and LFA1 (*Wang et al., 2019*). It might be interesting, however, to assess if lymphoma patients that developed refractory tumors following administration of CAR T cells have increased cell-in-cell formation.

This work also highlights a theoretical limitation of anti-tumor immunity, as even in a model system where all the 'breaks' on T cell activity are removed, tumor cells may activate a STAT3 transcriptional program, which promotes spatial architecture impenetrable to cytotoxic mediators. Indeed, over the past decade clinical studies tested various combinations of chemotherapies or tumor-inhibiting drugs with immunotherapy, especially in melanoma and NSCLC (*Apetoh et al., 2015*; *Shafique and Tanvetyanon, 2019*). However, since most tumor-inhibiting drugs also affect T cell activity, what is the optimal combination and sequence has been difficult to determine (*Karachaliou et al., 2017*). As a result, many of these trials have been using immunotherapy followed by chemotherapy. Here, we suggest that immune-mediated entosis should be inhibited prior to immunotherapy in order to reduce tumor resistance to subsequent drugs.

Overall, the ability of tumor cells to transiently enter and disseminate from each other in response to T-cell killing is a biological process that has never been described heretofore. It better explains how immunogenic tumors can survive in the host and provides a novel framework for immunotherapies.

## Materials and methods

### Mice

WT C57BL/6 and BALB/cOlaHsd mice were purchased from Envigo Israel. NSG (NOD/SCID-gamma KO mice) mice were a kind gift from Professor Michael Milyavsky at Tel Aviv University. Splenocytes Perforin KO mice (C57BL/6-Prf1tm1Sdz/J) were a kind gift from Professor Wolf-Dietrich Hardt at Institute of Microbiology, ETH Zurich. All mice were housed in an American Association for the Accreditation of Laboratory Animal Care–accredited animal facility and maintained under specific pathogen-free conditions. Animal experiments were approved and conducted in accordance with Tel-Aviv University Laboratory Accreditation #01-16-095, #01-21-011, and #01-19-034. Male and female 8- to 12-week-old mice were used in all experiments.

### Tumor models

For melanoma tumor studies, $2.5\times10^5$ B16F10 cells in 50 µL DMEM were injected s.c. into C57BL/6 mice above the right flank. For a triple-negative breast cancer model, $2\times10^5$ 4T1 cells in 30 µL DMEM were injected into mammary fat pad number five. Tumor size was measured twice a week using calipers. For immunotherapy, tumor-bearing mice were injected intratumorally twice, 2 days apart, with 100 µg anti-CD40 (FGK4.5; BioXCell), 0.2 µg TNFα (BioLegend), and 200 µg/mouse anti-TRP1 antibody (TA99; BioXCell). For adoptive cell transfer, splenic T cells were infected with pMIGII encoding TCR recognizing MHCI-gp100$_{25-33}$(*von Boehmer et al., 2013*), or MHCI-TRP2$_{180-188}$(*Tran et al., 2016*), or MHCII-TRP1$_{113-126}$(*Yost et al., 2019*). Recipient mice were sub-lethally irradiated at a single dose of 600 rad, and injected i.v. twice, 3 days apart, with $1 \times 10^6$ transduced T cells followed by i.p. injections of 300,000 IU of IL2 (PeproTech) for 4 consecutive days.

### Cell lines

B16F10 cells (CRL-6475) and 4T1 (CRL-2539) cells were purchased directly from ATCC in January 2017 and used no later than passage four. A375 and MDA-MB-231 cell lines were a kind gift from Professor

Tamar Geiger from Tel Aviv University. Both authentications were performed at the Genomics Core Facility of BioRap Technologies and the Rappaport Research Institute in Technion, Israel. Short tandem repeat profiles were determined using the Promega PowerPlex 16 HS Kit. HEK-293FT were purchased from Thermo Fisher Scientific in January 2017. EPH4 cells were a kind gift from professor Neta Erez from Tel Aviv University. Cells were cultured in DMEM (Gibco, Thermo Fisher Scientific) or RPMI1640 (Biological Industries) supplemented with 10% heat-inactivated FBS (Biological Industries), 2 mM L-glutamine, and 100 µg/mL penicillin/streptomycin (Gibco, Thermo Fisher Scientific) under standard conditions. Melan-A cells were purchased from Welcome Trust Functional Genomics Cell Bank in February 2019 and were supplemented with 200 nmol/L of phorbol 12-myristate 13-acetate (Santa Cruz Biotechnology). Cells were routinely tested for mycoplasma using a EZ-PCR Mycoplasma Test Kit (Biological Industries, Israel) according to the manufacturer's instructions.

## Antibody-mediated immunotherapy
Tumor-bearing mice were injected intratumorally twice, 2 days apart, with 100 µg anti-CD40 (clone FGK4.5; BioXCell), 0.2 µg TNFα (BioLegend), and 200 µg/mouse anti-TRP1 antibody (clone TA99; BioXCell).

## Primary immune cell isolation
All tissue preparations were performed simultaneously from each individual mouse (after euthanasia by $CO_2$ inhalation). Spleen was homogenized in 2% FBS and 5 µmol/L EDTA supplemented HBSS through 70 µm strainer (Thermo Fisher Scientific). Tumors were digested in RPMI 1640 with 2 mg/mL collagenase IV, 2000 U/mL DNase I (both from Sigma-Aldrich) for 40 min at 37 °C with magnetic stirrers (200 rpm). Cells were then washed by centrifugation at 600 rcf for 5 min at 4–8°C. Single-cell suspension from tumor and spleens was applied of a Histopaque-1077 Hybri-Max (Sigma-Aldrich) gradient and centrifuge for 15 min at 600 rcf.

For T-cell isolation, splenic mononuclear cells were incubated with anti-CD4 or anti-CD8 magnetic beads (MojoSort Nanobeads, BioLegend) according to the manufacturer's instruction.

For NK isolation, splenic mononuclear cells were incubated with anti-CD4, anti-CD8, anti-B220, anti-CD19 and anti-CD115 magnetic beads (MojoSort Nanobeads, BioLegend) according to the manufacturer's instruction, and collected through negative selection.

Macrophages were isolated from peritoneal cavity of euthanized mice by washing and recollecting 5 ml of HBSS twice, followed by centrifugation at 600 rcf.

Serum was acquired by pulling blood from venae cava of euthanized mice. Blood was incubated on ice for 30 min and centrifugation of at 400 rcf for 10 min. Serum was collected and recentrifuged at 21,000 rfc for 10 additional minutes.

## Patients' samples
Paraffin block of primary and involved lymph nodes from six stage IV melanoma patients, who has gotten no treatment, were sectioned at 4 µM and stained with H&E according to established protocol. Additionally, paraffin block of primary and recurrent from untreated stage-I melanoma patient were sectioned at 4 µM and stained with H&E. In addition, we have isolated the tumor cells and T cells from freshly resected tumors obtained from two stage IIb melanoma patients.

## T-cell culture and expansion
T cells were cultured in RPMI 1640 supplemented with 1% Pen-Strep, 10% heat-inactivated FBS, 1% Sodium pyruvate, 1% MEM-Eagle non-essential amino acids, 1% Insulin-Transferrin-Selenium (Pepro-Tech, Rocky Hill, NJ), 50 µM β-mercaptoethanol (Sigma-Aldrich, Merck, Israel) and, unless mentioned otherwise, were supplemented with 1000 IU/mL recombinant murine IL-2 (PeproTech, Rocky Hill, NJ) on tissue culture plates pre-coated with 0.5 µg/mL anti-CD3 antibodies (clone 17A2, Biolegend).

## Tumor cell lentiviral transduction
For preparation of lentivirus, $1.3×10^6$ HEK-293FT cells were plated on a six-well plate precoated with 200 µg/mL poly-L-lysine and let to adhere overnight. pLVX plasmids containing H2B-GFP, H2B-tdTomato, MyrPalm-tdTomato (*Zacharias et al., 2002*), LifeAct-GFP, Wasabi or tdTomato under EF1 promoter together were mixed with psPAX2 (a gift from Didier Trono, Ecole Polytechnique Tédérale

de Lausanne, Lausanne, Switzerland; Addgene plasmid 12260) and pCMV-VSV-G (a gift from Bob Weinberg, Massachusetts Institute of Technology, Cambridge, Massachusetts, USA; Addgene plasmid 8454) at a molar ratio of 3:2:1, and cells were transfected using Polyplus jetPRIME reagent (Polyplus Transfection). After 24 hr, medium was replaced with complete DMEM supplemented with 0.075% sodium bicarbonate. Medium-containing viruses were collected after 24 hr and 48 hr.

For tumor cell infection, virus-containing media were mixed with 100 μg/mL polybrene (Sigma-Aldrich) and added to B16F10 cells ($1 \times 10^6$ cells on 35 mm tissue culture) for 30 min at 37 °C 5% $CO_2$. Cells were centrifuged for 30 min at 37 °C and 450 *g*. Then, 80% of the medium was replaced with complete DMEM supplemented with 0.075% sodium bicarbonate. After 3 days in culture, cells were sorted by FACSAriaIII. Cells were tested for mycoplasma, endotoxins, and bacterial contamination.

## T cells retrovirus Infection

HEK-293FT (Invitrogen) were plated on 10 cm culture plates and co-transfected with a 2:1 molar ratio of pMIGII and PCL-Eco plasmids (both were provided by Dario A.A. Vignali, University of Pittsburgh) using Polyplus jetPRIME reagent (Polyplus Transfection). After 24 hr, the medium was replaced with complete DMEM supplemented with 0.075% sodium bicarbonate. Media-containing viruses were collected after 24 hours and 48 hr and centrifuged for 1 hr at 100,000 *g*. The pellet was resuspended gently in 1 mL media and let to recover overnight at 4 °C.

Prior to infection, splenic T cells were incubated on a plate precoated with anti-CD3 (0.5 μg/mL) in T cell medium containing high-dose IL-2 (1000 IU/mL). Next, 0.3 mL concentrated retroviruses were added to every group of $2\times10^6$ cells with 10 μg/mL polybrene. Cells were incubated for 30 min at 37 °C in 5% $CO_2$ and centrifuged at 37 °C, 600 *g*, for 1 hr. Afterwards, 80% of medium was replaced and T cells were cultured for an additional 3 days in T cell media containing 1000 IU IL-2. Transduction efficacy was assessed by FACS as the percentages of GFP-expressing cells.

## Adoptive T-cells Transfer

T cells were isolated from spleens of naïve mice as specified above and infected with pMIGII encoding TCR recognizing MHCI-gp100$_{25-33}$ (*Overwijk et al., 1998*), or MHCI-TRP2$_{180-188}$ (*Bendle et al., 2010*), or MHCII-TRP1$_{113-126\ (61)}$. Recipient tumo-bearing mice were sub-lethally irradiated at a single dose of 600 rad using cobalt source. After 24 hr, mice were injected i.v. twice, 3 days apart, with $1 \times 10^6$ transduced T cells followed by i.p. injections of 300,000 IU of IL2 (PeproTech) for 4 consecutive days.

## Flow cytometry

For cell surface staining, monoclonal antibodies conjugated to FITC, PE, PE-Cy7, PE-Cy5.5, APC-Cy7, eFluor 650, or Pacific Blue and specific for the following antigens were used: CD11b (M1/70), Gr-1 (RB6-8C5), F4/80 (BM8), B220 (RA3-6B2), CD45 (A20), CD3 (17A2), CD4 (RM4-4), CD8 (53.6.7), MHCK$^b$/K$^d$ (28-8-6), CD115 (AFS98), I-Ab (M5/114.15.2), from BioLegend (San Diego, CA). Flow cytometry was performed on CytoFLEX (Beckman Coulter) and datasets were analyzed using FlowJo software (Tree Star, Inc). All in vivo experiments to characterize tumor-infiltrating leukocytes were independently repeated at least three times with three to five mice per group. iTAg APC-labeled H-2K$^b$-Trp-2(SVYDFFVWL) and iTAg PE-labeled H-2D$^b$-gp100(EGSRNQDWL) tetramers were purchased from MBL international (Woburn, MA) and were used according to manufacturer's instruction. Tetramers-staining experiments were repeated twice with 5 mice in each group.

## Measuring caspase 3/7 activity in tumor cells

For apoptotic cells measurements, $1 \times 10^4$ fluorescently-labeled tumor cells were seeded on 96-well round bottom plates and let adhere for several hours. T cells were then added to the culture in ratios of 1:1, 1:2, 1:5, and 1:10 for additional 3, 6, or 18 hr before adding CellEvent Caspase-3/7 Green Detection Reagent (Invitrogen). Fluorescent intensity was measured by Flow cytometry, using Cyto-FLEX (Beckman Coulter) and datasets were analysed using FlowJo software (Tree Star, Inc).

## Confocal microscopy

For confocal microscopy imaging, cells were cultured on glass-bottom confocal plates (Cellvis). Cells were live-imaged or fixed and permeabilized with 2% PFA for 20 min. Fixed cultures were washed twice with PBS and stained for membranes with DiD membrane dye (Invitrogen), F-actin with Alexa

Fluor 594 Phalloidin (Invitrogen) and nuclei with Hoechst 33,342 (Fluka). For immunofluorescence, fixed cultures were blocked overnight with 5% BSA and stained with 1:100 or 1:200 diluted primary antibodies. Cells were then washed with PBS containing 1% BSA and stained with secondary antibody, diluted 1:100 or 1:200. We used the primary antibodies: anti-CD44 (clone IM7), anti-MHC class I (clone 28-8-6)(both from BioLegend), anti-β-catenin (clone D2U8Y), anti-phosphorylated STAT3 (clone D3A7), anti-active integrin 1β (polyclonal), anti-Granzyme B (clone 5A1E)(all from Cell Signaling Technology), anti-integrin 1β (clone 9EG7), anti-E-cadherin (clone 36)(both from BD Biosciences), anti-TRP2 (clone C-9, Santa Cruz Biotechnology), anti-gp100 (clone EP4863 *Galon et al., 2006*) and anti-phosphorylated EGFR (clone EP38Y)(both from Abcam).

For frozen sections, tumor tissues were fixed in 4% paraformaldehyde for two hours and equilibrated in a 20% sucrose solution overnight. Tissues were then embedded in frozen tissue matrix (Scigen O.C.T. Compound Cryostat Embedding Medium, Thermo Fisher Scientific) and frizzed at –80 °C. All specimens were imaged by ZEISS LSM800 (ZEISS) and analysed by ZEN 2.3 (ZEISS) and Imagej.

## Image analysis

Multi-Z-stack images were taken using ZEISS LSM800 (ZEISS) confocal microscope.

Mean fluorescence intensity (MFI) was calculated in regions of interest (ROI) using ZEN 2.3 software (ZEISS) out of a set of images taken under the same conditions. For time-lapse quantification, MFI was normalized using MIN/MAX normalization: $norm\ X = \frac{X - min(X)}{max(X)\ - min(X)}$.

Three-dimensional projection and rendering and surface detection were done using IMARIS software 9.5 (Bitplan, RRID: SCR_007370), using the software's default parameters.

## Electron microscopy

Scanning Electron Microscopy preparation was done as described by Fischer et al, with minor modifications (*Fischer et al., 2012*). Briefly, cells were fixed with warm 2.5% glutaraldehyde (GA) (EMS) in PBS for 60 min. Cells were then washed twice for 5 min in PBS, and twice with cacodylate buffer (0.1MCaCO, 5mMCaCl pH7.3) (Merck), post-fixed with 1% $OsO_4$ for 60 minutes, washed three times in cacodylate buffer and then twice with $H_2O$; Dehydration was done with increasing concentrations of reagents grade ethanol (2×5 minutes for 25%, 50%, 70%, 95% and 2×10 min for 100%), and was followed by coating with gold nanoparticles. Images were captured using GeminiSEM 300 (Carl Zeiss Microscopy, Jena, Thuringia, Germany).

For Transmitting Electron Microscopy cells were fixed in 2.5% Glutaraldehyde in PBS over night at 4 °C. After several washings in PBS cells were post fixed in 1% OsO4 in PBS for 2 hr at 4 °C. Dehydration was carried out in graded ethanol followed by embedding in Glycid ether. Thin sections were mounted on Formvar/Carbon coated grids, stained with uranyl acetate and lead citrate and examined in Jeol 1400 – Plus transmission electron microscope (Jeol, Japan). Images were captured using SIS Megaview III and iTEM the Tem imaging platform (Olympus).

## Incucyte assays

For long-term live cell imaging, B16F10 expressing H2B- TdTomato, H2B-GFP, and a mixture of both were sorted from immunotherapy-treated mice and plated in flat-bottom 96-well tissue culture (Corning) and placed in IncuCyte S3 Live Cell Analysis System (Essen BioScience).

For viability assays, $1 \times 10^4$ B16F10 cells expressing H2B-TdTomato were plated in flat-bottom 96-well tissue culture ($10 \times 10^3$ per well, Corning) and were let to adhere for several hours. Splenic $CD8^+$ T cells infected with TCR against gp100, or TRP2 were added to culture at several ratios ranging from 1:1 to 10:1 E:T and placed in IncuCyte S3 Live Cell Analysis System (Essen BioScience). In other experiments, $1 \times 10^4$ B16 cells expressing H2B-TdTomato were plated in flat-bottom 96-well tissue culture (Corning) and incubated in full DMEM medium containing either 1.5 mM doxorubicin, $3 \times 10^{-5}$% $H_2O_2$ or 50 mg/ml cycloheximide for 92 hr. Confluence percentage and images were acquired and analyzed using IncuCyte Base Analysis Software (Essen BioScience).

Relative confluence was calculated by normalizing to confluence on $T_1$:

$$relative\ con\ Tn = \frac{con\ Tn}{con\ T1}$$

## Neoantigen predication

Genomic DNA was extracted from $5 \times 10^6$ B16F10 tumor cell lines using NucleoSpin Tissue (MACHEREY-NAGEL), according to manufacturer instructions. Whole exome sequencing was performed by NovoGene. Sequencing libraries were constructed using the Agilent SureSelect kit and sequence on an Illumina Novaseq 6,000 sequencer. Sequencing reads were filtered using the AGeNT SureCall Trimmer (Agilent, v4.0.1) and quality was evaluated using FastQC v0.11.3. Alignment to the mouse genome (mm10, GRCm38.87) was performed using the Burrows-Wheeler Aligner (bwa v 0.7.12) using the bwa mem command with the following parameters: -M –t 4 –T 30. SAMtools was used to remove alignments with a MAPQ score less than 5, and Picard was used to remove duplicate reads. Sorting and indexing were performed using SAMtools. Variants were called using MuTect2 (GATK v 4.0.0.0) and filtered using FilterMutectCalls. Variants were annotated with SnpEff. Neoantigens were identified using MuPeXI v1.2.0 (*Bjerregaard et al., 2017*) with the following parameters: -s mouse -a H-2-Db, H-2-Kb -l 9–11. MuPeXI was used in conjunction with NetMHCpan v.4.0 and Variant Effect Predictor (VEP) v.87.27. Heatmaps were generated in R using the 'pheatmap' package (v1.0.12).

## mRNA extraction and RNAseq analyses

To extract mRNA from tumor cells, $5 \times 10^6$ B16F10 sorted from in vivo relapsed tumors, or from cell B16F10 cell lines established from relapsed tumors were collected. For preparation cell-in-cell culture induced by T cells-derived secreted granules, $1 \times 10^6$ B16F10 cells were seeded on a low-adherence 10 cm plastic plate pre-coated with 200 µg/mL poly-L-lysine and were let to adhere for 3 hr. Concentrated T cell–derived secreted granules in DMEM were added to the culture, in 1:10 ratio for 4 hr. Cells were collected by washing with fresh media followed by 5 min centrifugation at 250 *g*. RNA was extracted from cells using NucleoSpin RNA (MACHEREY-NAGEL), according to manufacturer instructions. Sequencing Libraries were prepared using INCPM mRNA Seq. NextSeq 75 cycles reads were sequenced on 8 lanes of an Illumina nextseq. The output was ~18 million reads per sample.

For B16 RNAseq, sequencing adaptors were trimmed using fastp 0.19.6 (*Chen et al., 2018*) and aligned to the GRCm38 mouse assembly using STAR 2.7.1 a (*Dobin et al., 2013*). Gene count matrix was analyzed for differential expression using DESeq2 1.24.0 (66). Enrichment analysis was performed on differentially expressed genes (adjusted p-value < 0.05) using clusterProfiler 3.12.0. (*Yu et al., 2012*).

For T cells RNAseq, reads were aligned using Kallisto (*Bray et al., 2016*) to mouse genome version mm10, followed by further processing using the Bioconductor package DESeq2 1.24.0 (*Love et al., 2014*). The data was normalized using TMM normalization, and differentially expressed genes were defined using the differential expression pipeline on the raw counts with a single call to the function DESeq (FDR- adjusted p-value < 0.05). Heatmap figures were generated using pheatmap package (*Kolde and Vilo, 2015*) and clustered using Euclidian distance. Boxplots were generated using BoxPlotR. Sequencing data was deposited in GEO under accession codes GSE164076, GSE164075 and GSE164077.

## Cell-derived secreted granules and melanosomes isolation

For isolation of T-cell–derived secreted granules, $2 \times 10^7$ CD8$^+$ T cells were isolated from spleen and plated in 10 cm plate precoated with anti-CD3 and anti-CD28 antibodies and high-dose IL-2 (1,000 IU/ mL) for 48 hours in T cell media containing 10% FBS pre-centrifuged at 140,000 rpm for one hour to deplete bovine-derived exosomes. Supernatants were collected and concentrated using Amicon Ultra (Merk) with 100kD filter. For isolation of NK cells-derived secreted granules, splenic $3 \times 10^6$ NK cells isolated from naïve mice were plated in 12-well plate incubated with high-dose IL-2 for 48 hr. Supernatants were collected and concentrated using Amicon Ultra (Merk) with 100kD filter. Secreted granules were added to B16F10 cultures for 24–48 hr incubation and the numbers of cell-in-cell structures out of total 200 cells were counted under fluorescent microscope.

For isolation of secreted melanosomes from B16F10, Melan A, $8 \times 10^6$ cell were cultured in 10 cm tissue culture plates and let to adhere overnight. Cells were incubated 48 hr in DMEM supplemented with 10% exosome-free FBS. Supernatants was collected and centrifuged for 1 hr on glucose gradient, as previously described (*Santana-Magal et al., 2020*).

## Cell-in-cell inhibition assays

For inhibition assays, B16F10 were plated in 4 Chamber glass-bottom confocal plates (Cellvis) 3000 cells/chamber and let adhere for several hours. Cells were incubated for 24–48 hr in complete DMEM, with or without 50 µg/mL cycloheximide, 120 mM 5,6-dichloro-1-beta-D-ribofuranosylbenz imidazole (DRB) (Sigma-Aldrich), 200 nM Latrunculin B (Lat-B, Sigma-Aldrich), 2 µM Stattic (Sigma-Aldrich), 4 µM Gefitinib, 5 µM XAV-939 (Sigma-Aldrich) or the blocking antibodies anti-E-cadherin (clone 36) and anti-N-cadherin (clone 13A9). In addition, gp100-reactive T cells (E:T ratio of 5:1), or T cells-derived secreted granules (10–50 µL of concentrated secreted granules per well) were added to the culture. Cell-in-cell structures were counted under fluorescent microscope for cell-in-cell per 200 cells, three times for each chamber.

## Cell-in-cell counts

For cell-in-cell count, 3000–10,000 tumor cells per one $cm^2$ were plated in a 4 Chamber glass-bottom confocal plates (Cellvis) or 96-well flat bottom (Corning) and let adhere for several hours. Cells were then incubated overnight with immune cells or exosomes. Cells were then washed with PBS, trypsinized and moved to coverslip for counting under fluorescent microscope. Percentages of cell-in-cells structures were calculated by counting 200 cells in three independent samples, three fields for each sample.

## Study approval

All animal protocols were approved by the Tel-Aviv University Institutional Animal Care and Use Committee under protocol: #01-16-095, #01-21-011, and #01-19-034. The Rabin Medical Center and Tel Aviv University Institutional Review Board approved the human subject protocols 0460–19-RMC and 16–660-TLV-7 respectively, and informed consent was obtained from all subjects prior to participation in the study.

## Statistics

For time course experiments, significance was calculated using the nonparametric two-way ANOVA with Tukey's correction for multiple hypotheses. In some cases, Bonferroni-Sidak post-test was performed after two-way ANOVA. For two groups analysis, one-way ANOVA with Dunn's test was performed. The results were analyzed by Prism (GraphPad Software, Inc). All statistical analyses were performed in Prism (GraphPad Software, Inc). All experiments were performed at least three times with at least three replications.

## Acknowledgements

The authors would like to deeply thank Rabin Medical Center Institutional Tissue Bank and especially Dr. Adva Levi-Barda, for their invaluable support of this research. We would also like to thank Dr. Vered Holdengreber from the Faculty of Life Sciences, Tel Aviv University and Dr. Gal Radovsky from Tel Aviv University Center for Nano Science and Nano Technology for their help with electron microscopy imaging.

## Additional information

### Funding

| Funder | Grant reference number | Author |
| --- | --- | --- |
| Fritz Thyssen Stiftung | | Yaron Carmi |
| Israel Cancer Research Fund | | Yaron Carmi |
| Israel Science Foundation | | Yaron Carmi |
| Tel Aviv University | | Yaron Carmi |

| Funder | Grant reference number | Author |
|---|---|---|

The funders had no role in study design, data collection and interpretation, or the decision to submit the work for publication.

## Author contributions

Amit Gutwillig, Conceptualization, Investigation, Writing - original draft; Nadine Santana-Magal, Leen Farhat-Younis, Diana Rasoulouniriana, Investigation; Asaf Madi, Software, Formal analysis; Chen Luxenburg, Validation, Methodology; Jonathan Cohen, Investigation, Methodology; Krishnanand Padmanabhan, Resources, Investigation, Methodology; Noam Shomron, Guy Shapira, Nathan Edward Reticker-Flynn, Software, Formal analysis, Methodology; Annette Gleiberman, Investigation, Writing - review and editing; Roma Parikh, Resources, Formal analysis, Investigation, Methodology; Carmit Levy, Reno Debets, Resources, Methodology; Meora Feinmesser, Resources, Validation, Methodology; Dov Hershkovitz, Valentina Zemser-Werner, Oran Zlotnik, Sanne Kroon, Wolf-Dietrich Hardt, Resources; Peleg Rider, Conceptualization, Investigation, Methodology, Writing - original draft; Yaron Carmi, Conceptualization, Supervision, Investigation, Methodology, Writing - original draft

## Author ORCIDs

Asaf Madi ⓘ http://orcid.org/0000-0003-3441-3228
Guy Shapira ⓘ http://orcid.org/0000-0001-9376-4955
Annette Gleiberman ⓘ http://orcid.org/0000-0002-4155-4224
Sanne Kroon ⓘ http://orcid.org/0000-0002-5722-0763
Wolf-Dietrich Hardt ⓘ http://orcid.org/0000-0002-9892-6420
Reno Debets ⓘ http://orcid.org/0000-0002-3649-807X
Yaron Carmi ⓘ http://orcid.org/0000-0002-0972-0616

## Ethics

Human subjects: The Rabin Medical Center and Tel Aviv University Institutional Review Board approved the human subject protocols 0460-19-RMC and 16-660-TLV-7 respectively, and informed consent was obtained from all subjects prior to participation in the study.

All mice were housed in an American Association for the Accreditation of Laboratory Animal Care-accredited animal facility and maintained under specific pathogen-free conditions. All animal protocols were approved by the Tel-Aviv University Institutional Animal Care and Use Committee under protocols: #01-16-095, #01-21-011, and #01-19-034. Male and female 8-12 week-old mice were used in all experiments.

## Decision letter and Author response

Decision letter https://doi.org/10.7554/eLife.80315.sa1
Author response https://doi.org/10.7554/eLife.80315.sa2

# Additional files

## Supplementary files
• MDAR checklist

• Source data 1. Western blot images of phosphorylated genes in cells from cell-in-cell tumor formations, related to *Figure 6*.

## Data availability
Sequencing data have been deposited in GEO under accession codes GSE164076, GSE164075 and GSE164077.

The following datasets were generated:

| Author(s) | Year | Dataset title | Dataset URL | Database and Identifier |
|---|---|---|---|---|
| Gutwillig A, Santana-Magal N, Farhat-Younis L, Rasoulouniriana D, Madi A, Luxenburg C, Cohen J, Padmanabhan K, Shomron N, Shapira G, Feinmesser M, Zlotnik O, Debets R, Reticker-Flynn NE, Rider P, Carmi Y | 2020 | Transient cell-in-cell formation underlies tumor relapse and resistance to immunotherapy [T cells] | https://www.ncbi.nlm.nih.gov/geo/query/acc.cgi?acc=GSE164076 | NCBI Gene Expression Omnibus, GSE164076 |
| Gutwillig A, Santana-Magal N, Farhat-Younis L, Rasoulouniriana D, Madi A, Luxenburg C, Cohen J, Padmanabhan K, Shomron N, Shapira G, Feinmesser M, Zlotnik O, Debets R, Reticker-Flynn NE, Rider P, Carmi Y | 2020 | Transient cell-in-cell formation underlies tumor relapse and resistance to immunotherapy [B16F10] | https://www.ncbi.nlm.nih.gov/geo/query/acc.cgi?acc=GSE164075 | NCBI Gene Expression Omnibus, GSE164075 |
| Gutwillig A | 2020 | Transient cell-in-cell formation underlies tumor relapse and resistance to immunotherapy | https://www.ncbi.nlm.nih.gov/geo/query/acc.cgi?acc=GSE164077 | NCBI Gene Expression Omnibus, GSE164077 |

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
