## [Editor Report]

This is a timely and important study that describes a new potential mechanism of resistance to immune checkpoint blockade. Not only does this have significant implications for cancer immunotherapy, but could extend to other immunological malignancies as well.

---

## [Decision Letter]

**Decision letter after peer review:**

Thank you for submitting your article "Transient cell-in-cell formation underlies tumor relapse and resistance to immunotherapy" for consideration by *eLife*. Your article has been reviewed by 2 peer reviewers, one of whom is a member of our Board of Reviewing Editors, and the evaluation has been overseen by W Kimryn Rathmell as the Senior Editor. The reviewers have opted to remain anonymous.

Essential revisions:

1) Clarification of the treatments used and relevance to current immunotherapy practices. If the therapy used is just a "means to an end", this is ok but should be clearly stated as such.

2) More clarity in figures and legends.

3) Analysis of treated tumors for Figure 2I.

*Reviewer #1 (Recommendations for the authors):*

1. It is unclear what the tumor-binding antibody is and what is its mechanism. Is this a T cell engager? Meant to induce ADCC?

2. The treatment studied (anti-CD40+TNFa+tumor-binding antibodies) is largely irrelevant to most cancer therapies used today. Broadly linking observations with this therapeutic regimen to what is seen with conventional immunotherapy (i.e. checkpoint blockade) should be met with caution and should be more extensively discussed. The impact and relevance to translational cancer research would be much improved if this was studied during PD-1 blockade.

3. The figure legends are inadequate to fully understand what was done and what the figures are showing. For example, in Figure 2I, what is the difference between the left and right figures? What is the red stain vs. the green stain?

4. If Figure 2I is untreated tumors, this same analysis should be shown for treated tumors.

5. These structures don't look like cells inside of other cells, therefore the term cell-in-cell is misleading.

---

## [Author Response]

Reviewer #1 (Recommendations for the authors):1. It is unclear what the tumor-binding antibody is and what is its mechanism. Is this a T cell engager? Meant to induce ADCC?

We apologize for not making this issue clearer. Briefly, we have previously demonstrated that in order to elicit DC-mediated T cell immunity using tumor-binding antibodies, they must be delivered along with DC adjuvant (e.g. TLR agonist such as LPS, CpG, or a combination of cytokines such TNF and anti-CD40). These findings and the underlying molecular mechanisms were published in Nature 2015 (25924063) and JCI insight 2016 (27812544). We added a section to the Introduction clarifying the rational of choosing this treatment.

2. The treatment studied (anti-CD40+TNFa+tumor-binding antibodies) is largely irrelevant to most cancer therapies used today. Broadly linking observations with this therapeutic regimen to what is seen with conventional immunotherapy (i.e. checkpoint blockade) should be met with caution and should be more extensively discussed. The impact and relevance to translational cancer research would be much improved if this was studied during PD-1 blockade.

This is a fair point. Unfortunately, the effect of classical checkpoint blockade therapy in animal models is limited to a small number of benign tumors and it is challenging to employ it for relapsed tumor models. We used a DC-based immunotherapy that is shown to induce tumor-eradicating T cell immunity (Carmi Y. *Nature* 2015, 25924063; Spitzer M. and Carmi Y. *Cell* 2017, 28111070; Ackerman S. *Nature Cancer* 2022, 35121890) and its potency in human tumors is not been tested in multi-center clinical trials by Bolt Biotherapeutics. To test the relevance of our findings to additional immunotherapies, we also used adoptive transfer of tumor-reactive CD8^+^ T cells as a treatment for melanoma-bearing mice, which is a common practice in treating melanoma. With that said, we have toned down some of the language and conclusions. Additionally, we have added sentences to the Discussion that highlight the limitation of our experimental model.

3. The figure legends are inadequate to fully understand what was done and what the figures are showing. For example, in Figure 2I, what is the difference between the left and right figures? What is the red stain vs. the green stain?

We thank the reviewer for pointing that out. We have revised the figure legends across the manuscript to make them more understandable. As for Figure 2I, we replaced it to better present the morphological differences in tumor cells following T cell derived immunotherapy.

4. If Figure 2I is untreated tumors, this same analysis should be shown for treated tumors.

We have replaced Figure 2I. It now compares histological sections from untreated and immunotherapy-treated mice.

5. These structures don't look like cells inside of other cells, therefore the term cell-in-cell is misleading.

This is a fair point. We changed the language to describe more accurately these structures. They are now referred as structures where one or more nuclei were surrounded by multiple membranes.